# Spatiotemporal behavior of the far wake of a wind turbine model subjected to harmonic motions: phase averaging applied to stereo particle image velocimetry measurements

**Antonin Hubert, Boris Conan, and Sandrine Aubrun**

CE1 Nantes Université, École Centrale Nantes, CNRS, LHEEA, UMR 6598, 44000 Nantes, France

**Correspondence:** Sandrine Aubrun (sandrine.aubrun@ec-nantes.fr)

**Abstract.** The complex dynamics introduced by floating platforms present new challenges in the study of wind turbine wakes, and numerous questions remain unresolved due to the early stage of this technology and the limited operational experience. Some previous studies showed that harmonic motions with realistic amplitude and frequency under a modeled atmospheric boundary layer have no significant impact on time-averaged values due to the relatively high background turbulence, but they also show that frequency signatures are still visible in spectra of wake parameters. The purpose of this work is to shed light on the spatiotemporal behavior of the wake imposed by surge, heave, and pitch harmonic motions. Wind tunnel experiments on the wake of a porous disk immersed in a modeled marine atmospheric boundary layer were performed, and a phase-averaging method with kernel smoothing was applied to the data to extract the harmonic response of the wake. A quasi-steady-state analysis was carried out, showing that the phase-averaged observations appear to be larger than simple steady-wake model predictions and revealing the dynamic nature of the wake responses to the motion. Thus, distinct wake dynamic hypotheses are formulated depending on the nature of the motion: (i) for heave, the wake is translated vertically while maintaining its integrity and containing the same power; (ii) for surge, the wake contracts and expands without any displacement of its center localization, accompanied by in-phase power modulation; (iii) and for pitch, the wake dynamics include both heave and surge impacts, with a vertical translation of the wake synchronized with crosswise wake surface and power modulations.

## 1 Introduction

In the context of growing global energy demand, floating offshore wind turbine (FOWT) technology shows great potential. Unlike onshore, the offshore environment offers substantial advantages such as unobstructed wind flow and stronger winds, resulting in a higher capacity factor for installed wind turbines. The increasing distance from the coast provides access to stronger and more constant winds, which boost the wind turbine productivity from a capacity factor of 30 %–35 % for new onshore installations to 42 %–55 % for new offshore ones (Costanzo et al., 2023). Over the past decades, research on bottom-fixed wind turbines has led to a significant

advance in the comprehension of their wake (Ainslie, 1988; Vermeer et al., 2003; Larsen et al., 2007; Porté-Agel et al., 2020). However, the intricate dynamics motion introduced by floating platforms poses new challenges with respect to wind turbine wakes, and several questions remain unanswered due to the infancy of this technology and the lack of feedback: the first pilot floating farm, made of five turbines, has only been in operation since 2017 in Scotland (Hywind Scotland).

The floating motions depend on the platform technology itself (spar, tri-floater, barge, tension leg platform, etc.), but the most widely studied motions are surge and pitch – i.e., fore–aft translation and rotation – as they represent the common case of aligned wind–waves (Porchetta et al.,

2019). However, other motions such as sway and roll (side to side translation and rotation) and heave (up-down translation) have also been studied in the literature. By defining the Strouhal number $St = \frac{f \cdot D}{U_{\text{hub}}}$ and the normalized amplitude $A^* = \frac{A}{D}$ where $A$ and $f$ are the motion amplitude and frequency, respectively; $D$ is the rotor diameter; and $U_{\text{hub}}$ is the velocity at hub height, FOWT motions can be divided into two types: the high-amplitude and low-frequency ones ($A^* \approx 0.1$, $St < 0.3$) relative to mooring lines (Leimeister et al., 2018) and the low-amplitude and high-frequency ones ($A^* \approx 0.01$, $St > 0.5$) typically caused by a linear response of the platform to wave solicitation (Feist et al., 2021).

Several numerical and experimental studies have investigated the effects of imposed or free motion on FOWT wake characteristics. Sebastian and Lackner (2013) showed that the different types of FOWT present significant unsteady aerodynamic loading for high Strouhal number motions ($St > 0.5$). Bayati et al. (2017) evaluated the impact of high Strouhal number surge motions with wind tunnel experiments using hot-wire anemometry and observed great unsteadiness and nonlinearity in the wake. Bayati et al. (2018) and Fontanella et al. (2021) showed that surge motions impact the tip vortex and by extension the near wake by adding energy to the wake. Other studies observed the impact of imposed motions in the near and far wake at lower frequencies of $St \in [0.0085, 0.28]$ CE2 in the case of low-turbulence flows (Rockel et al., 2014; Fu et al., 2019; Kopperstad et al., 2020; Meng et al., 2022). They observed faster wake recovery for imposed motion cases compared to the fixed one due to a larger shear layer between the wake and the freestream and increased turbulent kinetic energy (TKE) in the wake.

Kopperstad et al. (2020) investigated the wake of a wind turbine mounted on a barge and a spar using wind tunnel experiments and computational fluid dynamics (CFD) simulations for uniform low and high turbulent inflow conditions with realistic mooring line motions ($St < 0.25$). They showed that the higher pitch and surge platform motion amplitudes of the barge concept generate strong coherent flow structures and thus shear layer instabilities at the limit of the wake, resulting in faster wake recovery. More recently, Li et al. (2022) studied the wake of a FOWT subjected to sway and roll motions using large-eddy simulations and linear stability analysis in uniform and low-turbulent-inflow conditions (TI < 4 %). The results revealed that a turbine motion of $St \in [0.2, 0.6]$ can trigger large-scale far-wake meandering even at low amplitudes ($A^* = 0.01$).

Based on their numerical results, Kleine et al. (2022) concluded that motion impacts the tip vortices by exciting vortex instability modes almost identical to the motion itself. They noted that pitch motion shows a combination of heave and surge effects: vortices merge into one large structure that is coherent in the streamwise direction, the same as in the surge, but vortices in the lower vertical direction coalesce faster than the ones in the higher vertical direction, approach-

ing heave effects. Duan et al. (2022) observed that a wind turbine model under surge motion with $St = 0.55$ formed periodical vortex rings in its wake, which were not visible in surge motion with a higher Strouhal number. These effects were visible for all motion amplitudes tested, ranging from $0.01D$ to $0.06D$.

Messmer et al. (2024) studied a reduced-scale wind turbine under surge and sway motions in wind tunnel experiments under low-turbulence-intensity inflow conditions (TI ≈ 0.3 %). They showed that motion leads to faster wake recovery for sway motions in a range of $St \in [0.3, 0.6]$ and for surge motions in a range of $St \in [0.3, 0.9]$. Like Li et al. (2022), they found that sway motions result in quasi-periodic meandering phenomena, while surge motions lead to streamwise pulsation in the wake for $St \in [0.25, 0.5]$ and lateral meandering for $St \in [0.5, 0.9]$. Using a different approach, Li and Yang (2024) computed numerical simulations of a turbine subjected to surge motions with much higher Strouhal numbers ($A^* = [0.11, 0.44]$, $St = [3.5, 14]$) and under different turbulent inflow conditions. Thus, they observed periodic coherent structures induced in the wake by the surge motions at both turbulence levels. Moreover, these structures appear to enhance the advection process, making the wake recovery faster.

Fu et al. (2023) computed numerical simulations of a reduced-scale model of a FOWT subjected to pitch motion of different motion amplitudes (1, 4, and 10°) and with a high Strouhal number ($St = 1.1$). They showed that the relative velocity seen by the rotor is modified, impacting the power and loads applied to the turbine, and that the higher the motion amplitude, the greater its impact. Moreover, they found that the wake recovery is enhanced by motions caused by tip and root vortex instabilities. With the objective of predicting the wake characteristics of a FOWT, Zhang et al. (2024) proposed an analytical wake model for a turbine subjected to pitch motions. After a validation with two wind tunnel experiments, with $St = 0.02$ and 0.06 and $A = 5.5°$ and $17.6°$, they found that the wake predictions vary with time, implying velocity modulations persisting in the far wake similar to those in previously cited studies of surge motions.

Regarding these studies and their results, three ranges of motion of the Strouhal number can be drawn according to their impacts on the wake. (i) The first one is motions at low Strouhal numbers ($St < [0.1, 0.2]$), where the wake is moved in a quasi-steady state succession. The motion impacts the global wake, and structures of the flow are displaced according to the turbine motion (e.g., Fu et al., 2019, 2020; Meng et al., 2022; Zhang et al., 2024). (ii) On the opposite side are motions at high Strouhal numbers ($St > [0.6, 0.7]$), where the motion impacts shear instabilities at the wake borders. The motion induces mild oscillations in the wake with a too high frequency compared to its characteristic aerodynamic timescale (e.g., Fu et al., 2023; Messmer et al., 2024; Li and Yang, 2024). (iii) Finally, between these two ranges are motions where they develop wake dynamics (e.g., Kopperstad

et al., 2020; Fontanella et al., 2021; Li et al., 2022; Kleine et al., 2022; Messmer et al., 2024; Schliffke et al., 2024).

The majority of previous studies were performed with low levels of turbulence and no shear inflow conditions. How-
[5] ever, shear, higher turbulence intensity, and spectral contents developed by realistic atmospheric conditions may impact the wake differently and lead to other conclusions. FOWT wake studies under realistic inflow conditions are necessary to provide information on the impact of platform motions and
[10] to improve dynamic wake models in the context of floating wind farms. However, realistic atmospheric turbulent conditions lead to additional challenges. Turbulent structures contained in the approaching flow at a scale larger than the disk diameter are responsible for the appearance of wake me-
[15] andering, a low-frequency nonharmonic phenomenon characterized by a global displacement of the wake (Larsen et al., 2007; España et al., 2011). This phenomenon should not be confused with the motion-induced wake meandering observed in previously cited studies; wake meandering signifies
[20] a displacement of the global wake in a crosswise direction, but this can be caused by the turbulent large-scale structures present in the inflow – thus appearing in the wake of both bottom-fixed and floating wind turbines – or be caused by the motion of the floating platform – thus only appearing in
[25] the wake of the FOWT.

Because of wake meandering, the wake location fluctuates in the crosswise direction. This generally results in a time-averaged wake that is larger than the instantaneous one, and the wake radius or velocity deficit can be misjudged. To solve
[30] this issue, two reference frames are used to process the turbine wake statistics (Bingöl et al., 2009; Larsen et al., 2019; Jézéquel et al., 2022): (i) the fixed frame of reference (FFoR), where the analysis is computed within the fixed reference of the experiments, and (ii) the moving frame of reference
[35] (MFoR), where the wake statistics are analyzed relative to the wake center, preventing the spread of statistics caused by meandering.

The majority of experimental studies are performed with rotating wind turbine models, but the complexity of their de-
[40] sign related to the difficulty of reproducing the right aerodynamic loads, especially under realistic turbulent conditions caused by the absence of Reynolds similarity, leads to the use of porous disk models (España et al., 2011; Aubrun et al., 2013). Due to the absence of tip vortices and angular momen-
[45] tum, a porous disk can be considered a far-wake generator, focusing the problem on the far-wake instabilities rather than on any other sources of instabilities. The goal of a porous disk is to reproduce the pressure difference found throughout the rotor of a real wind turbine. The turbulence and the
[50] rotational momentum created by each individual blade are not reproduced by this model, but it has been proven that these structures are negligible in the far wake (Vermeer et al., 2003) – i.e., from $3D$ to $4D$ downstream of the wind turbine – and that a three-blade rotating wind turbine and a porous
[55] disk present the same wake properties even under low turbu-

lence intensity ($I_u = 4\%$ after a downstream distance of $3D$; Aubrun et al., 2013).

Using the atmospheric boundary layer (ABL) wind tunnel facility of the LHEEA (Research Laboratory in Hydro-
[60] dynamics, Energetics and Atmospheric Environment,École Centrale de Nantes), Belvasi et al. (2022) and Schliffke et al. (2024) performed hot-wire and stereo particle image velocimetry (S-PIV) measurements with a porous disk subjected to low-Strouhal-number heave, surge, and pitch mo-
[65] tions ($St \in [0.13, 0.38]$) under realistic turbulent inflow conditions. In these wind tunnel experiments $4.6D$ and $8.125D$ downstream of the FOWT model, they observed clear, unique signatures of the harmonic motion frequencies in the spectra of the wake parameters, such as the wake center or the available power. These signatures are amplified with higher-
[70] Strouhal-number motion, showing a 4-fold energy spectrum amplitude increase from a surge motion of $St = 0.25$ to $St = 0.35$. However they showed that because of the high level of turbulence, the shear layer, and the presence of meandering due to the ABL modeling, the conventional time-averaged
[75] results are not appropriate for distinguishing differences between cases with and without motions and cannot be used to observe the actual motion impact on the wake. The investigations revealed that despite the frequency signatures, in similar motion conditions the imposed motions produce no
[80] discernible effect on velocity statistics, including streamwise velocity, its standard deviation, and the turbulent kinetic energy. Additionally, the statistics of wake meandering, such as the standard deviation and the skewness and kurtosis of the wake center coordinates, demonstrated no clear sensitivity to
[85] these imposed motions.

The aim of the present paper is to extend the previous work performed under realistic inflow conditions by studying the spatiotemporal wake behavior and to check whether the findings obtained at low turbulence intensity and uni-
[90] form inflow still hold. In the present article, the position, surface, and available power of the far wake of a porous disk are investigated by extracting the averaged phase associated with the harmonic motions studied in their time series. For this purpose, S-PIV experiments are conducted down-
[95] wind of the porous disk subjected to heave, surge, and pitch harmonic motions and under a neutral marine ABL modeled at a reduced scale of $1 : 500$. The unsteady wake properties are described in FFoR and in MFoR given the presence of wake meandering, and a phase-averaged method with kernel
[100] smoothing is applied to the resulting velocity fields.

Section 2 details the experimental methodology and the data postprocessing, including the wake center computation, essential for the MFoR, and the explanations of the phase-averaging method. The computational procedure and nota-
[105] tions employed in this study are also presented in this section. Section 3 displays the phase-averaged results, and Sect. 4 presents the analysis of the results in three parts, corresponding to the different types of motion imposed. Finally, a conclusion is given in Sect. 5.
[110]

## 2 Methodology

### 2.1 Experimental setup

#### 2.1.1 Atmospheric boundary layer physical modeling

Experiments were conducted in the atmospheric boundary layer wind tunnel of the LHEEA – Research Laboratory in Hydrodynamics, Energetics and Atmospheric Environment – at École Centrale de Nantes in France. It is a $2\,\text{m} \times 2\,\text{m}$ cross-section and 24 m long facility, with a 45 kW motor allowing a maximum flow velocity of $10\,\text{m s}^{-1}$. As described by Schliffke et al. (2024), a $1:500$ neutral marine ABL is modeled using a trip and spires installed at the entrance of the test section and using perforated metal plates placed on the ground, as shown in Fig. 1. The boundary layer develops along a 20 m long fetch before reaching the current properties.

The resulting velocity profile corresponds to a neutral ABL developing on slightly rough terrain according to VDI (2000), with a roughness length of $z_0 = 5.7 \times 10^{-3}$ m at full scale, a power-law exponent of $\alpha = 0.11$, and a zero-plane displacement of $d_0 = 0$ m. The integral length scale of turbulence is about ${}^x L_u = 240$ m at the hub height, while the target one is ${}^x L_u = 250$ m according to Counihan (1975). The ABL parametrization and its dependence on the type of terrain have been largely validated through observational statistics (Kaimal and Finnigan, 1994; Counihan, 1975) and have led to guidelines on the physical modeling of such an ABL in a wind tunnel (VDI, 2000). Nevertheless, the potential modification of the marine ABL according to the sea state is disregarded in the present study; the complexity of the wind–wave–wake interactions are not fully modeled and can impact the observed results (Porchetta et al., 2019, 2021; Ferčák et al., 2021).

Figure 2 shows the normalized longitudinal mean velocity $U_{\text{ABL}}/U_{\text{hub}}$, with $U_{\text{hub}} = 2.9\,\text{m s}^{-1}$ as the velocity at the hub height (Reynolds number of $Re = 3 \times 10^4$) and the three components of the turbulence intensity measured in the wind tunnel defined by the formula $I_i = \frac{\sigma_i}{U_{\text{ABL}}}$, $i = u$, $v$, or $w$. More details on the turbulence and on the ABL modeling are presented in Schliffke et al. (2024). Moreover, this same study showed that the assumption of Reynolds number independence is valid.

#### 2.1.2 Model description and test conditions

The model used in the wind tunnel is based on a $1:500$ reduced-scale model of a 2 MW floating wind turbine (80 m in diameter, 60 m at the hub height, as detailed by Choisnet, 2013). In the test section, this wind turbine is modeled with a porous disk with a diameter of $D = 160$ mm, a hub height of $z_{\text{hub}} = 120$ mm, and a surface of $S_{\text{disk}} = \pi \frac{D^2}{4} = 0.020\,\text{m}^2$, which gives a blockage ratio of 0.5 % in the test section. It has a solidity of $\sigma = 57\%$, which is slightly below the limit at which vortex shedding can appear in the wake; a thrust coefficient of $C_T = 0.65$; and a representative power coefficient of $C_P = 0.25$ according to Aubrun et al. (2019). Figure 3 shows the six degrees of freedom of a FOWT and their definitions.

The motion amplitudes and frequencies of a barge-type platform were extracted from a database of numerical simulations provided by the company BW-Ideol and are specific to the second-order motions related to the mooring lines and anchors acting on the floating platform – the first order being related to the response of the floater to wave-to-wave solicitations (Schliffke et al., 2024). They were converted into values at a reduced scale using kinematic similarity, resulting in velocity-scale and timescale factors between the full and reduced scales of 2.5 and 200, respectively. More details are available in Schliffke et al. (2024). Porous disk motions are imposed by a three-degrees-of-freedom system, which allows floating wind turbine movements in the $(x, z)$ plane. This study considers one heave case ($T_z$), one surge case ($T_x$), and two pitch cases ($R_y$) in addition to the one reference fixed case.

All motion cases, as detailed in Table 1, are defined by a sinusoidal function along time $t$:

$$A_{\text{motion}}(t) = A \sin(2\pi f t). \tag{1}$$

Full-scale configurations were downscaled to wind tunnel configurations by conserving the same normalized amplitudes and Strouhal numbers of the motions; amplitudes are normalized by the diameter. The pitch motion has a rotation center located at the floater level and can be considered a combination of tilt (pitch with a rotation axis at the disk center), surge, and heave motions: the 4° amplitude corresponds to an 8.4 mm amplitude surge with a 0.3 mm amplitude heave, as visible in Fig. 4.

#### 2.1.3 Stereo-PIV system

An S-PIV system, as represented in Fig. 1, is used to measure the three velocity components in the plan normal to the flow $(y, z)$ at $x = 8.125D$ downstream of the turbine model. This value corresponds that used in the previous experiments done by Schliffke (2022), Belvasi et al. (2022), and Schliffke et al. (2024) to observe FOWT wake dynamics. Moreover, this $8.125D$ value is realistic compared to the full-scale distances between two wind turbines in a wind farm (Dalla Longa et al., 2018). The flow is seeded with olive oil droplets with a diameter of 1 µm, sprayed by a LaVision seeding system. The laser system is a Nd-YAG double-cavity laser ($2 \times 200$ mJ) emitting two pulses with a wavelength of 532 nm and a thickness of approximately 3 mm at a time delay of 350 µs, with an emission rate set to 14.1 Hz, avoiding phase-locking with the motions.

The velocity measurement uncertainty in S-PIV systems is a combination of the numerous uncertainties present in the measurement chain and is related to the installation and to the postprocessing algorithms (Raffel et al., 1998; Wieneke,

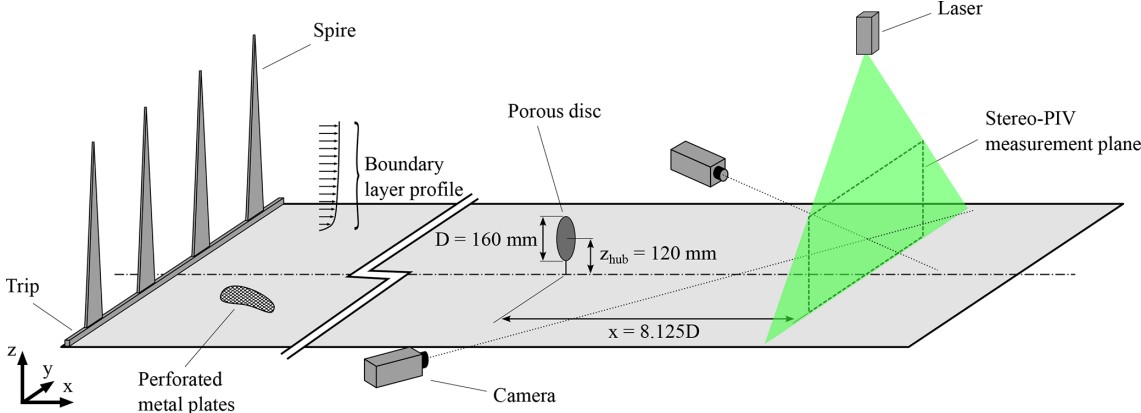

**Figure 1.** Experimental setup in the atmospheric boundary layer wind tunnel at École Centrale de Nantes.

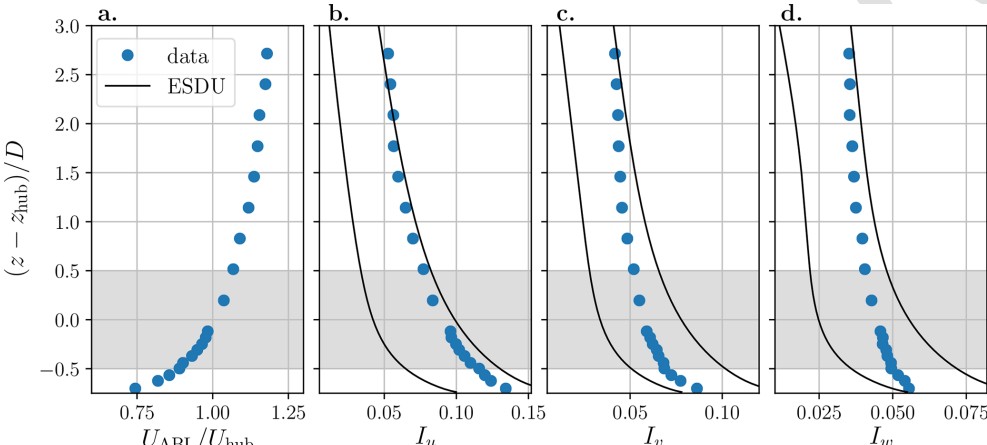

**Figure 2.** Normalized mean streamwise velocity profiles in the wind tunnel **(a)** and turbulence intensity profiles in the streamwise **(b)**, transversal **(c)**, and vertical **(d)** directions. The gray areas represent the turbine model height range, and the black lines represent the range of turbulence intensity expected for slightly rough terrain (ESDU, 1985). Figure from Schliffke et al. (2024).

2017; Sciacchitano, 2019). Adrian and Westerweel (2011) stated that a typical value of the S-PIV measurement uncertainty displacement of the particles is 0.1 pixel units. However, this is highly simplistic and should be treated with caution since, as mentioned earlier, the uncertainties vary according to the experimental setup.

For each configuration, 14 000 image pairs are acquired by two sCMOS 5.5 Mpx HighSense Zyla cameras with 60 mm Nikon objective lenses located on each side of the test section, as presented in Fig. 1. The mean field is subtracted to remove the persistent background, and the resulting images are processed with a three-pass adaptive correlation from a $128\,\text{px} \times 128\,\text{px}$ to a $32\,\text{px} \times 32\,\text{px}$ interrogation window size with an overlap of 50 %. Finally, the two-component vector fields are combined to reconstruct the instantaneous three-component velocity field.

## 2.2 Data processing

### 2.2.1 Wake center tracking

In order to compute the statistics in the MFoR, the instantaneous wake center is identified using the weighted geometric center (WGC) method with an exponential weighting of the velocity deficit. WGC is a method that has already been used in the literature by Muller et al. (2015), with an exponential weighting, or by Howland et al. (2016) without one.

Figure 5 illustrates the algorithm: each instantaneous S-PIV vector field is computed as a velocity deficit field by removing the time-averaged freestream velocity field $U_{\text{ABL}}$ (Fig. 5a), and the field is smoothed using a Gaussian filter to reduce the influence of the local turbulence (Fig. 5b). The Gaussian filter is defined as

$$f_{\text{G}}(y, z) = \frac{1}{2\pi\sigma^2} \exp\left(-\frac{y^2 + z^2}{2\sigma^2}\right), \qquad (2)$$

where $\sigma = 0.26D$ is the variance of the Gaussian function.

**Table 1.** Parameters of the motions imposed on the wind turbine model.TS1

| Motion | Full-scale amplitude | Full-scale period [s] | Model-scale amplitude | Model-scale frequency [Hz] | Normalized amplitude | Strouhal number |
|---|---|---|---|---|---|---|
| Fixed | – | – | – | – | – | – |
| Heave **H** | 2.5 m | 133 | 5 mm | 1.5 | 0.03 | 0.09 |
| Surge **S** | 5 m | 100 | 10 mm | 2 | 0.06 | 0.11 |
| Pitch **P$_{0.14}$** | 4° | 80 | 4° | 2.5 | 4°CE3 | 0.14 |
| Pitch **P$_{0.28}$** | 4° | 40 | 4° | 5 | 4° | 0.28 |

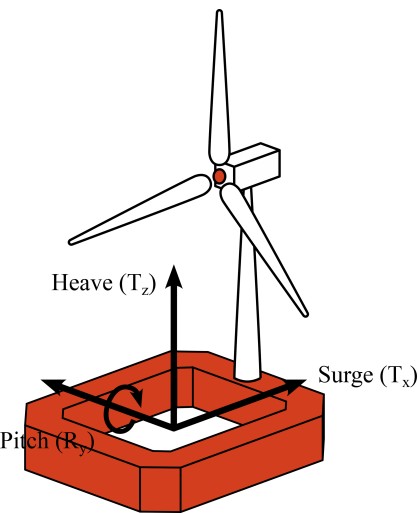

**Figure 3.** Degrees of freedom of a floating wind turbine platform. Surge ($T_x$) and heave ($T_z$) are the translation motions studied, and pitch ($R_y$) is the rotation motion studied.

A binarization process is then performed using a threshold velocity deficit set to $U_{\text{thresh}} = 0.1 U_{\text{hub}}$ to determine the points that are part of the wake (Fig. 5c). This process consists of pixel detection, inspired by watershed processing (Beucher, 2004) where each pixel verifies if its neighboring pixels belong to the wake by comparing the local velocity deficit value to the threshold one. As in España et al. (2011), the wake is unified to decrease pixel noise (Fig. 5d), and the final field provides the integration surface $S_{\text{wk}}$ to be used in the WGC to find the wake center coordinates $y_c$ and $z_c$:

$$\left( y_c(t) = \frac{\iint\limits_{S_{\text{wk}}} y e^{\Delta u(t,y,z)} \mathrm{d}y\mathrm{d}z}{\iint\limits_{S_{\text{wk}}} e^{\Delta u(t,y,z)} \mathrm{d}y\mathrm{d}z}; \quad z_c(t) = \frac{\iint\limits_{S_{\text{wk}}} z e^{\Delta u(t,y,z)} \mathrm{d}y\mathrm{d}z}{\iint\limits_{S_{\text{wk}}} e^{\Delta u(t,y,z)} \mathrm{d}y\mathrm{d}z} \right), \quad (3)$$

where $\Delta u(t,y,z) = U_{\text{ABL}}(y,z) - u(t,y,z)$, $u(t,y,z)$ is the instantaneous velocity, and $U_{\text{ABL}}(y,z)$ is the mean inflow velocity at point $(y,z)$. $U_{\text{ABL}}$ is calculated for each case as the average of the velocity profiles of the mean field at the S-PIV measurement plane limits, which present the lowest porous disk impact.

The results show that the lower part of the wake is truncated, as in Fig. 5, due to the S-PIV measurement plane definition. This truncation could potentially misrepresent the wake surface but also the wake center coordinates, especially for $z_c$. As the wake descends, a new portion of the wake disappears under the S-PIV measurement plane, which leads to an "artificial" decrease in the wake surface and an increase in the wake center $z$ coordinate. Tests with an ideal Gaussian wake showed a difference of $0.1D$ between the WGC result and the real one when 20 % of the wake is cut off, representative of the worst case here – i.e., when the porous disk is at the bottom. Thus, considering the up–down motions (mainly pitch motion cases), the amplitudes of the wake statistics in MFoR are misrepresented. The consequences for the analysis of the curve trends are limited, however.

In order to reduce this effect for the phase-averaged wake centers, a Gaussian fit approach is used. In this method, a least-squared error method is computed between the S-PIV $u$-component velocity field and a 2D Gaussian function defined by

$$f_{\text{Gf}}(y,z) = A \exp\left[ -\frac{1}{2}\left( \frac{(y-y_c)^2}{\sigma_y^2} + \frac{(z-z_c)^2}{\sigma_z^2} \right) \right], \quad (4)$$

where $A$ is the amplitude, and $\sigma_y^2$ and $\sigma_z^2$ are the variances of the Gaussian function in the $y$ and $z$ directions, respectively. The wake center $(y_c, z_c)$ found is the location of the center of the 2D Gaussian function closest to the velocity field.

With a high level of turbulence, the instantaneous fields cannot be assimilated to a Gaussian distribution, and Gaussian fitting results in incoherent wake center values. Thus, the WGC method is applied to the instantaneous velocity fields and Gaussian fitting to the phase-averaged ones (the processes are detailed in Fig. 8).

### 2.2.2 Phase-averaging and kernel smoothing

The phase-averaging method is applied to the S-PIV velocity fields according to the harmonic motion imposed on the porous disk in FFoR and in MFoR. A kernel smoothing, defined by an Epanechnikov function (Wand and Jones, 1995; Hastie et al., 2009), is used to smooth the velocity deficit fields over phases. The Epanechnikov kernel smoother is defined as

$$K_h(t_\phi, t) = \frac{3}{4h}\left( 1 - \left( \frac{t_\phi - t}{h\lambda} \right)^2 \right), \quad (5)$$

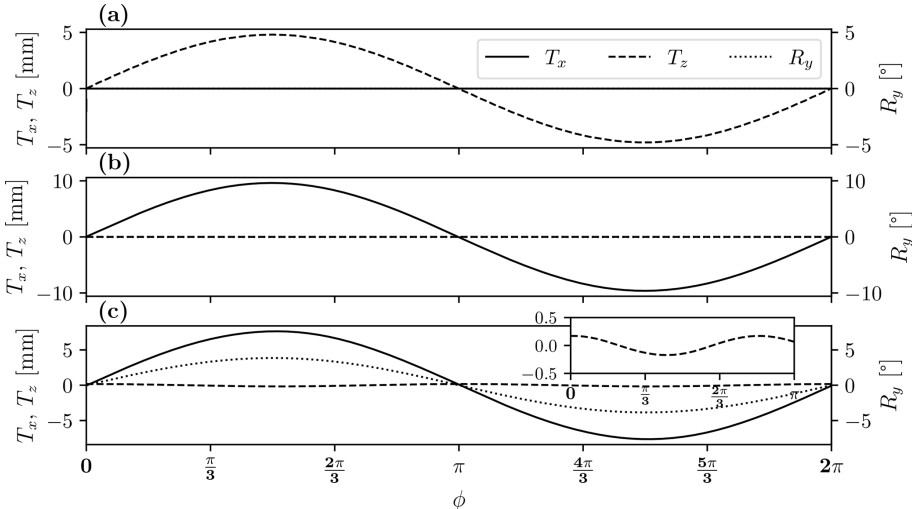

**Figure 4.** Imposed motion amplitudes for heave **(a)**, surge **(b)**, and pitch **(c)**. For heave and surge, the $R_y$ curve is blended with the $T_x$ and $T_z$ curves, respectively.

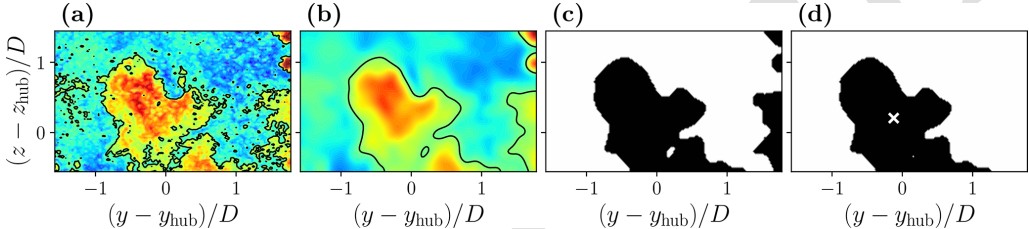

**Figure 5.** Presentation of the WGC algorithm steps: the instantaneous velocity deficit field (black lines represent the threshold value contour) **(a)**, Gaussian filtering **(b)**, image binarization **(c)**, and unification of the wake **(d)**.

where $t$ is the time; $t_\phi$ is the reference time corresponding to the function maximum $x$ axis; and $h$ and $\lambda$ are the length scale and the bandwidth, also called the window width, of the function, respectively.

Figure 6 illustrates the algorithm using the example of a noisy sinusoidal function of period $T$. Each sample $m(t)$ (Fig. 6a) is associated with its respective reference time phase $t_\phi$ according to a phase $\phi$ of the imposed harmonic motion (Fig. 6b), and the Epanechnikov kernel smoother $K_h$ is computed for each phase increment to obtain the phase-averaged curve $\widetilde{m}$ (Fig. 6c):

$$\widetilde{m}(t_\phi) = \frac{\sum\limits_{i=1}^{N} K_h(t_\phi, t_i) m(t_i)}{\sum\limits_{i=1}^{N} K_h(t_\phi, t_i)}, \tag{6}$$

where $N$ is the total number of data points in the defined interval of the Epanechnikov function at the reference time $t_\phi$ (corresponding to the gray zone in Fig. 6c). Using a confidence interval of 95 %, the statistical uncertainties, $I_{\widetilde{m}}$, are calculated using the formula

$$I_{\widetilde{m}}(t_\phi) = \left[ \widetilde{m}(t_\phi) - 2\frac{\sigma_{\widetilde{m}}(t_\phi)}{N}; \quad \widetilde{m}(t_\phi) + 2\frac{\sigma_{\widetilde{m}}(t_\phi)}{N} \right], \tag{7}$$

with $\sigma_{\widetilde{m}}$ as the standard deviation of $\widetilde{m}$,

$$\sigma_{\widetilde{m}}(t_\phi) = \frac{\sum\limits_{i=1}^{N} K_h(t_\phi, t_i) m(t_i)^2}{\sum\limits_{i=1}^{N} K_h(t_\phi, t_i)} - \widetilde{m}(t_\phi)^2. \tag{8}$$

Theoretically, the Epanechnikov function is an optimal kernel: the AMISE (asymptotic mean integrated square error) criterion of this kernel, which defines the global error in the function, is minimized regardless of the sample size compared to other forms such as the Gaussian or uniform ones (Wand and Jones, 1995). Nevertheless, the resulting values of kernel smoothing must be taken with caution, as the method acts like a low-pass filter and tends to limit extreme phenomena. Also, if a kernel function is too narrow, the result is based on too few data points and gives too much weight to each particular piece of data, resulting in an under-smoothed

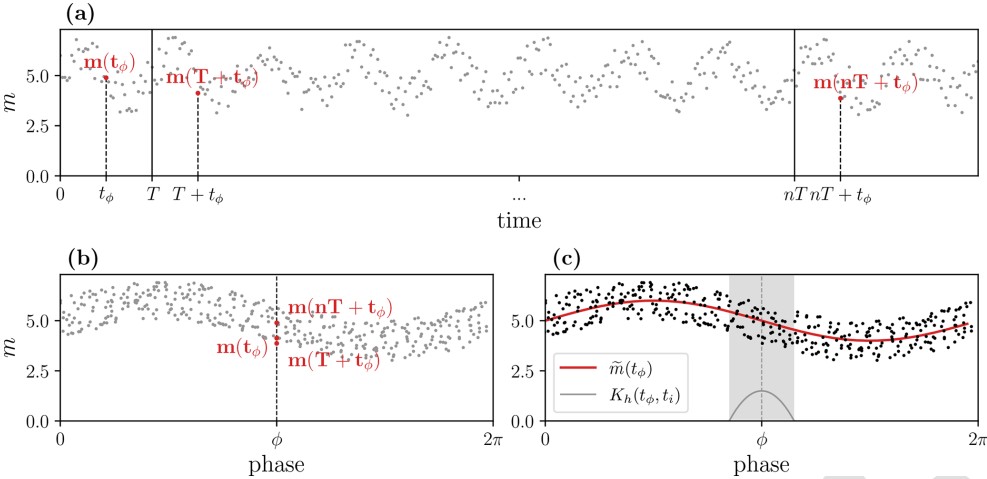

**Figure 6.** A phase averaging process example for a noisy sinusoidal function $m$ with a period of $T$: time series $m(t)$ **(a)**, phase association between data value $m(t)$ and the phase of the harmonic signal $\phi$ **(b)**, and phase-averaging with kernel smoothing (function $K_h(t_\phi, t_i)$) processing for each phase **(c)**. The gray area in **(c)** represents the $i$ points used to compute $\widetilde{m}(t_\phi)$.

estimation. Conversely, if the kernel function is too large, the result takes too many data points into account, resulting in an over-smoothed estimation. Thus, the choice of the bandwidth value $\lambda$ is important. Here, after an iterative study to prevent either under- or over-smoothed curves, the value was set equal to one-fifth of the total number of phases.

## 2.3  Frames of reference and wake metrics

Because of wake meandering due to large turbulent structures in the inflow and of wake movements induced by imposed motions, the results are analyzed from two perspectives: in the fixed frame of reference (FFoR), to observe the impact of the turbine motion on a potential downstream turbine, and in the moving frame of reference (MFoR), to study the wake under floating motions.

Figure 7 provides a comparison of the mean vertical and lateral velocity deficit profiles for the fixed case in both frames of reference. In the vertical direction (Fig. 7a), the profiles show similar shapes, but the maxima of the deficit are vertically shifted by $-0.12D$ and $-0.2D$ from the wake center in FFoR and MFoR, respectively. This is partly due to the investigation area, which truncates the wake at the bottom, causing the WGC algorithm to artificially shift the wake center higher in the vertical direction, as explained in Sect. 2.2.2. Another explanation is the presence of the ground, which causes the wake to become asymmetrical by distorting its lower part and which moves the maximum of the velocity deficit away from the actual wake center. This is discernible from the profile in FFoR: it plunges toward the lower velocity deficit values faster at the bottom than at the top, showing a velocity difference of $0.05U_{\text{hub}}$ reached with a height difference of $0.33D$ and $0.60D$ at the bottom and the top, respectively.

In the lateral direction (Fig. 7b) in either FFoR or MFoR, the velocity deficit profiles exhibit the same shape, with a maximum close to the wake center. Velocity deficit profiles in FFoR are flatter than those in MFoR. The difference reaches $0.04U_{\text{hub}}$ at the wake center and is nearly null at the extrema of the wake. Moreover, the velocity deficit profile in MFoR is positioned slightly to the right of the wake center, resulting in an intersection of the two profiles at $y - y_c = -0.62D$ and $y - y_c = 0.96D$. Lastly, the selection of FFoR or MFoR significantly influences the velocity deficit profiles. Figure 7 illustrates that without a preprocessing wake center tracking algorithm, there is a risk of underestimating the velocity deficit values, leading to an overestimation of the available power within the wake.

In this study, the wake parameters are computed according to the following procedure (illustrated in Fig. 8): (i) the mean inflow velocity field $U_{\text{ABL}}$ is subtracted from each instantaneous velocity field $u(t, y, z)$ to obtain the instantaneous velocity deficit field $\Delta u(t, y, z)$. (ii) The instantaneous wake center coordinates $(y_c(t), z_c(t))$ are computed using the WGC method described in Sect. 2.2.1 to obtain the velocity field in MFoR $u_m(t, y - y_c(t), z - z_c(t))$ and the velocity deficit field in MFoR $\Delta u_m(t, y - y_c(t), z - z_c(t))$ at each time step. (iii) Finally, the phase-averaging method with kernel smoothing, as detailed in Sect. 2.2.2, is applied to the four fields, resulting in the phase-averaged velocity field $\widetilde{u}(\phi, y, z)$ and velocity deficit field $\widetilde{\Delta u}(\phi, y, z)$ in FFoR and in the phase-averaged velocity field $\widetilde{u}_m(\phi, y - y_c(\phi), z - z_c(\phi))$ and velocity deficit field $\widetilde{\Delta u}_m(\phi, y - y_c(\phi), z - z_c(\phi))$ in MFoR.

From the fields in FFoR, for each phase $\phi$, the phase-averaged wake center coordinates $(\widetilde{y}_c(\phi), \widetilde{z}_c(\phi))$ calculated with the Gaussian fitting method described in Sect. 2.2.1 and

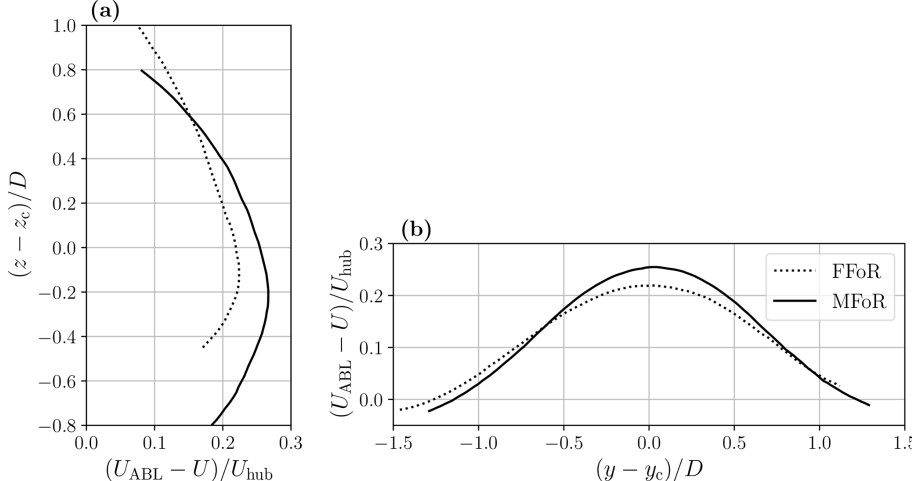

**Figure 7.** Mean velocity deficit profiles $U_{ABL} - U$ normalized by the velocity at hub level $U_{hub}$ for the fixed case in the vertical **(a)** and lateral **(b)** directions centered on the wake center coordinates $(y_c, z_c)$ in FFoR (dashed lines), calculated using the Gaussian fit method, and in MFoR (full lines).

the phase-averaged available power in FFoR defined as

$$\widetilde{P}(\phi) = \int_{S_{disk}} \frac{1}{2}\rho\widetilde{u}(\phi, y, z)^3 \mathrm{d}s \qquad (9)$$

are obtained, where $\rho$ is the air density integrated on a crosswise surface equal to the disk surface $S_{disk}$, which represents the potential wind power that a downstream turbine could produce. It is normalized by $P_{ABL}$, defined as the available power in the inflow integrated over the same surface:

$$P_{ABL} = \int_{S_{disk}} \frac{1}{2}\rho U_{ABL}(y, z)^3 \mathrm{d}s. \qquad (10)$$

From the fields in MFoR, for each phase $\phi$, the phase-averaged wake surface $\widetilde{S}_{wk}(\phi)$ defined by the integration surface used in the WGC method from Sect. 2.2.1 and the phase-averaged available power in MFoR defined as

$$\widetilde{P}_m(\phi) = \int_{S_{disk}} \frac{1}{2}\rho\widetilde{u}_m(\phi, y - y_c(\phi), z - z_c(\phi))^3 \mathrm{d}s \qquad (11)$$

are obtained. The integration is carried out on a crosswise surface equal to $S_{disk}$, and the result is normalized by $P_{ABL}$. $\widetilde{P}_m$ represents the actual power within the wake. With the WGC bias, considering the worst case – i.e., 20 % of the wake is truncated – the resulting $\widetilde{P}_m$ shows a relative error of about 14 %. Thus, for the heave and pitch motion cases, the analysis will essentially be performed on the curve trends and not on the amplitudes.

A summary of the wake parameters used in this work and of the necessary computational methods is presented in Fig. 8. $(y_c(\phi), z_c(\phi))$ are the wake center coordinates processed through the phase-averaged method applied to the velocity and velocity deficit fields in MFoR. They are not shown or used in the present work.

## 3 Results

This section provides the phase-averaged results for the different motion cases. A global description is presented here; analyses are presented in Sect. 4.

Figure 9 presents the phase-averaged velocity deficit fields. Each column corresponds to a motion case and each row to a motion phase. The phases are $[0, \frac{\pi}{3}, \frac{2\pi}{3}, \pi, \frac{4\pi}{3}, \frac{5\pi}{3}]$ CE4, corresponding approximatively to the phases when the wake parameter extrema appear. The dashed-line circle represents the fixed porous disk emplacement. The full black line is the velocity deficit contour of $U_{thresh}$, and the black cross is the wake center calculated from the phase-averaged velocity field. The gray lines and crosses represent the time-averaged wake contour and wake center, respectively.

In this figure, a preliminary analysis of the wake dynamics associated with the imposed motion can be conducted by comparing the black contours and crosses to the gray ones. In the fixed case (not shown here), the black contours and crosses coincide with the gray ones, indicating negligible velocity deficit modifications and the absence of periodic dynamics in this case. Therefore, any modifications in this figure, such as wake and wake center movement or modifications in velocity deficit values, can be attributed to the imposed harmonic motion.

Regarding the heave case (**H** – first column), the first fields depict the wake with an axisymmetric shape, and the wake center is located at the contour center (Fig. 9a and e). Then, the wake is distorted with a wider contour at the bottom, and the wake center descends to its lowest point (Fig. 9i). Following this phase, both the wake center and contour ascend, accompanied by an increase in their surface (Fig. 9m and q) until they reach their highest point (Fig. 9u). Moreover, the

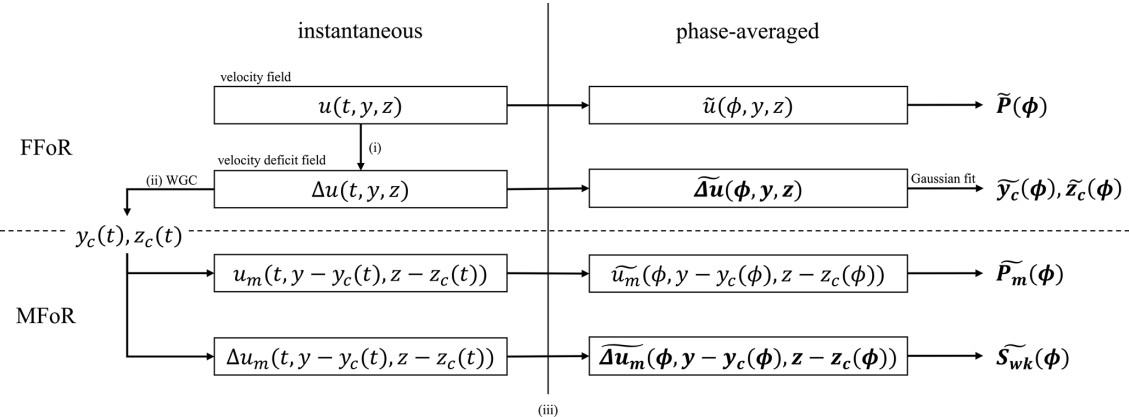

**Figure 8.** Schematic view of the wake parameters used in this work and their connections. The boxes show velocity fields, and bold variables are the parameters analyzed in this paper. The steps shown are (i) subtraction of the mean velocity field, (ii) calculation of the wake center positions with the WGC method, and (iii) phase averaging with kernel smoothing applied to the velocity fields.

velocity deficit values are modified across the phases, following the ascent and descent of the wake, with a maximum at phase $\frac{5\pi}{3}$ (Fig. 9u) and a minimum at phase $\frac{2\pi}{3}$ (Fig. 9i).

In the surge case (**S** – second column), the modifications are not as pronounced as for **H**. The wake center does not move significantly, and the wake contour maintains an axisymmetric shape throughout the phases. Nevertheless, there are notable modifications in the velocity deficit values, with a minimum amplitude at phase 0 (Fig. 9b) and a maximum at phase $\pi$ (Fig. 9n), greater than those for **H**.

Both pitch cases (**P$_{0.14}$** and **P$_{0.28}$** – last columns) exhibit similar wake dynamics to the heave case, involving ascending and a descending movement of the overall wake. **P$_{0.14}$** and **P$_{0.28}$** have a $\widetilde{z_c}$ minimum and maximum wake contour deformations at phase $\frac{\pi}{3}$ (Fig. 9g and h) and an increase in $\widetilde{z_c}$ and $\widetilde{S_{wk}}$ (Fig. 9k, l, o, and p) until a maximum at phase $\frac{4\pi}{3}$ (Fig. 9s and t), followed by their reduction to close the loop (Fig. 9w, x, c, and d). Additionally, the pitch cases exhibit modifications in the velocity deficit values similar to those in the surge case, with a minimum at phase $\frac{\pi}{3}$ (Fig. 9g and h) and a maximum at phase $\frac{4\pi}{3}$ (Fig. 9s and t). **P$_{0.28}$** shows more pronounced modifications with higher wake center and surface amplitudes, likely due to the higher wake receptivity, as observed by Schliffke et al. (2024).

Figure 10 presents the contours of the wake defined by the velocity deficit threshold value of $U_{thresh}$ and calculated in MFoR for different phases, where each row corresponds to a motion case. The interesting thing about this figure is that it enables the shape of the wake to be analyzed by suppressing its displacement effect. The center of each plot corresponds to the wake center for all phases. The evolution of the wake contours of **H**, **P$_{0.14}$**, and **P$_{0.28}$** consists of four distinct steps, as observed in the analysis of Fig. 9. Taking **P$_{0.28}$** as an example (Fig. 10d), at phase 0 the wake has an axisymmetric shape. Then, the wake widens in the negative $z$ area, resulting in a deformed shape from phase $\frac{\pi}{3}$ to $\frac{5\pi}{3}$. Its height increases

until it reaches the maximum of the wake surface at phase $\pi$ before regaining its axisymmetric shape while decreasing its surface until phase 0, closing the loop.

These distinct steps are also visible for **H** and **P$_{0.14}$**, with smaller modifications. Regarding **H**, it appears that the wake dynamics is shifted by an offset of $\frac{\pi}{3}$. The differences in the wake surface are notably pronounced at the bottom of the wake contour, where and when the wake is assumed to be flattened by the ground effects – i.e., from phase $\frac{\pi}{3}$ to $\pi$.

Globally, in the same way as for the FFoR velocity deficit fields (Fig. 9), the motion case **S** exhibits an axisymmetric shape throughout all phases in MFoR (Fig. 10b). The increase and decrease in the wake surface occur simultaneously in both the $y$ and $z$ directions but with a limited amplitude compared to the other motion cases. The surface reaches its minimum at phase 0 and its maximum at phase $\pi$.

Figure 11 shows the phase-averaged wake center coordinates ($\widetilde{y_c}$ $\widetilde{z_c}$) calculated with the WGC algorithm described in Sect. 2.2.1 and the mean fixed case values, which are depicted by the horizontal dashed lines. The colored zones representing the statistical uncertainties defined in Eq. (7) are not visible due to their small values.

Modifications of $\widetilde{y_c}$ are not significant, with peak-to-peak amplitudes ranging from $0.02D$ for **P$_{0.14}$** to $0.08D$ for **P$_{0.28}$**, whereas the peak-to-peak amplitudes of $\widetilde{z_c}$ are more pronounced, with $0.20D$, $0.12D$, and $0.21D$ for **H**, **P$_{0.14}$**, and **P$_{0.28}$**, respectively. The fact that all cases present slight modifications of $\widetilde{y_c}$ across the phases is certainly due to the motion directions investigated. Kleine et al. (2022) noted that motion impacts the wake, with perturbations similar to the nature of the motion. Here, the porous disk movements (heave, surge, and pitch) do not present any $y$-direction component, resulting in low perturbations in the $y$ direction. Moreover, **S** is the only motion with no movement in the $z$ direction, resulting in minimal $\widetilde{z_c}$ modifications.

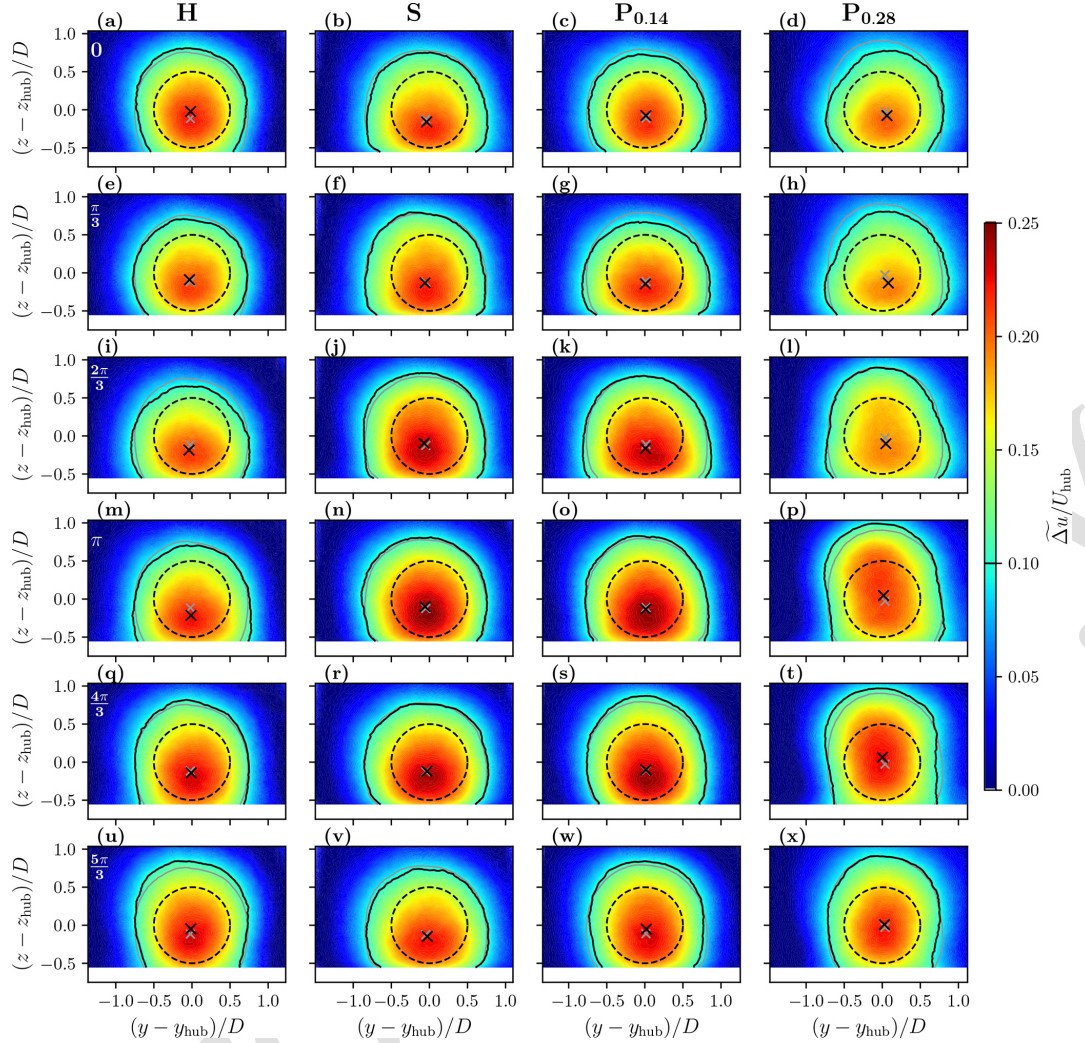

**Figure 9.** Phase-averaged velocity deficit fields $\widetilde{\Delta u}$ normalized by the inlet velocity at hub height for different phases $[0, \frac{\pi}{3}, \frac{2\pi}{3}, \pi, \frac{4\pi}{3}, \frac{5\pi}{3}]$ for **H**, **S**, **P$_{0.14}$**, and **P$_{0.28}$**. The dashed-line circle represents the porous disk emplacement; the black and gray full lines and crosses are the wake contours (0.1 normalized velocity deficit) and their centers, respectively (the black ones are phase averaged, while the gray ones are time averaged). The associated phases are visible in the upper-left corner of the images in the first row.

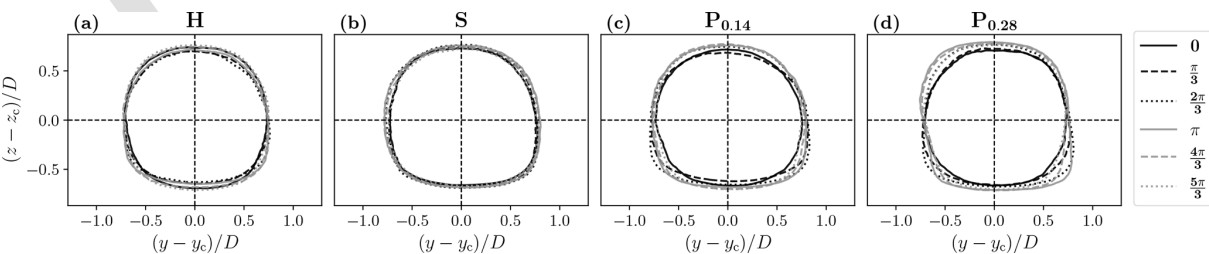

**Figure 10.** Phase-averaged wake contours in MFoR for different phases $[0, \frac{\pi}{3}, \frac{2\pi}{3}, \pi, \frac{4\pi}{3}, \frac{5\pi}{3}]$ for **H (a)**, **S (b)**, **P$_{0.14}$ (c)**, and **P$_{0.28}$ (d)**.

Figure 12 shows the phase-averaged wake surface $\widetilde{S_{\mathrm{wk}}}$ and the phase-averaged available power in FFoR $\widetilde{P}$ and in MFoR $\widetilde{P_{\mathrm{m}}}$, as defined in Sect. 2.3. $\widetilde{S_{\mathrm{wk}}}$ is computed in MFoR to avoid the overestimation present in FFoR for statistics in the presence of wake movement.

All motion cases exhibit clear variations in wake surface, with peak-to-peak amplitudes ranging from $0.48 S_{\mathrm{disk}}$ for **P$_{0.28}$** to $0.24 S_{\mathrm{disk}}$ for **S**. For the **S**, **P$_{0.14}$**, and **P$_{0.28}$** cases, these modifications are partially associated with changes in the velocity deficit values within the wake (Fig. 9). However,

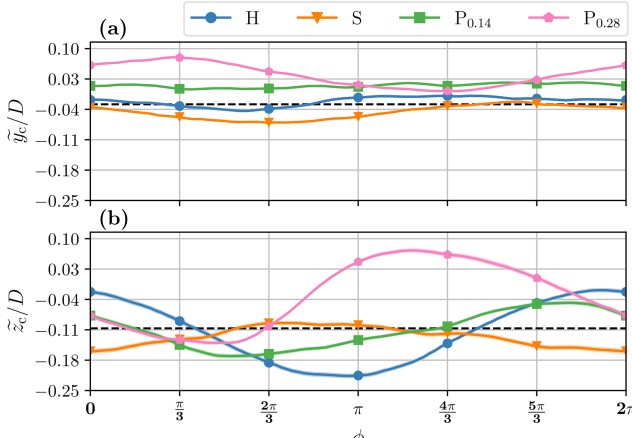

**Figure 11.** Phase-averaged wake center $y$ coordinates $\widetilde{y}_{\mathrm{c}}$ (a) and $z$ coordinates $\widetilde{z}_{\mathrm{c}}$ (b) for the different motion cases normalized by $D$. The colored zones represent the statistical uncertainties in the phase-averaged values defined in Eq. (7), and the horizontal dashed lines are the mean values for the fixed case. Each symbol corresponds to the phases already selected in the previous figures (Figs. 9 and 10).

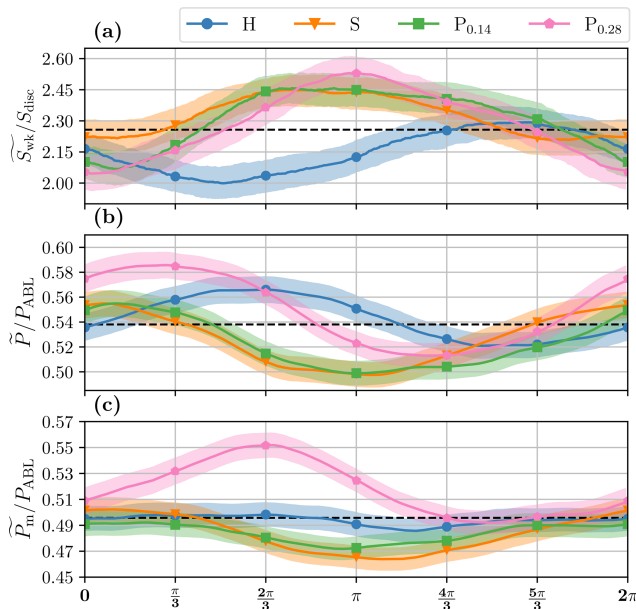

**Figure 12.** Phase-averaged wake surface $\widetilde{S_{\mathrm{wk}}}$ normalized by $S_{\mathrm{disk}}$ (a) and the available power in FFoR $\widetilde{P}$ (b) and in MFoR $\widetilde{P_{\mathrm{m}}}$ (c) normalized by the available power present in the ABL inflow $P_{\mathrm{ABL}}$ for the different motion cases. Same as Fig. 11.

as mentioned earlier in Sect. 2.3, the truncated part of the wake may lead to an overestimation of the actual wake surface variations. Indeed, the cases showing the greatest $\widetilde{S_{\mathrm{wk}}}$ modifications are those with the highest $\widetilde{z}_{\mathrm{c}}$ ones – i.e., **H**, **P₀.₁₄**, and **P₀.₂₈**. These motion cases present $\widetilde{S_{\mathrm{wk}}}$ variations similar to the worst case presented in Sect. 2.3 (20 % vs. 14 %, 17 %, and 21 % for **H**, **P₀.₁₄**, and **P₀.₂₈**, respectively).

All cases exhibit clear $\widetilde{P}$ variations, ranging from 8 % for **H** to 13 % for **P₀.₂₈**, compared to their mean values. These changes are inversely synchronized with $\widetilde{z}_{\mathrm{c}}$ for the **H**, **P₀.₁₄**, and **P₀.₂₈** cases – i.e., motions with a $z$-direction movement. Thus, in the same way as for $\widetilde{S_{\mathrm{wk}}}$, the $\widetilde{P}$ modifications may be partially attributed to the S-PIV measurement area.

As was said in Sect. 2.3, the analysis of $\widetilde{P_{\mathrm{m}}}$ was essentially performed on the curve trends and not on the amplitudes for the heave and pitch cases. For these cases, $\widetilde{P_{\mathrm{m}}}$ presents two different curve trends, with low variations of 3 % and 4 % for **H** and **P₀.₁₄**, respectively, compared to their mean values and higher ones of 8 % and 12 % for **S** and **P₀.₂₈**, respectively.

## 4  Discussion

In this section, the results from the preceding section are analyzed together, and the impacts of the motion on the porous disk wake are discussed for each type of movement successively: the study of the wake dynamics under heave motion (**H** – Sect. 4.1), surge motion (**S** – Sect. 4.2), and pitch motions (**P₀.₁₄** and **P₀.₂₈** – Sect. 4.3). Moreover, considering the low frequencies of the motions investigated ($St < 0.3$) in this section, the results are compared to simple steady-wake model predictions to evaluate whether the observed wake modifications are the result of the succession of steady states rather than of dynamic processes.

### 4.1  Heave motion

The present study of the heave motion effects suggests similarities with the LES (large-eddy-simulation) results from Li et al. (2022), manifesting a dynamic translation of the whole wake in the direction of the motion. Figure 13 shows a schematic view of the heave motion impact on the time evolution of the far wake of a FOWT, with wake center and wake surface modifications on the same order of magnitude as those found in the experiments. Heave motion causes the global wake to ascend and descend with the same period as the porous disk movement but with a higher amplitude, and it slightly modifies the intrinsic parameters of the wake, such as the wake surface or available power.

Across all phases, $\widetilde{y}_{\mathrm{c}}$ shows minimal variations (Figs. 11a and 12c). This is expected since heave motion imposes vertical movement on the porous disk. During heave motion, the porous disk moves in the shear layer and is therefore subjected to a hub velocity modification. However, this modification is negligible (less than 2 %; Fig. 2), which is consistent with the low variations in $\widetilde{P_{\mathrm{m}}}$.

The wake is moved vertically, with more than 3 times the amplitude of the disk motion: $0.1D$ for $\widetilde{z}_{\mathrm{c}}$ amplitude variations vs. $A_{\mathbf{H}} = 0.03D$. In order to check whether this discrepancy could be due to a wake deflection effect, a quasi-steady-state analysis was conducted. The vertical displacement of the porous disk creates an inflow skew, which deflects the wake in the opposite direction of the movement.

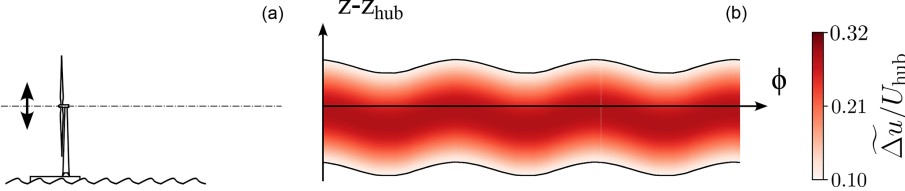

**Figure 13.** Schematic view of the heave motion impact on the wake based on the phase-averaged results. The FOWT is represented on the left **(a)** and the phase evolution of its far wake on the right **(b)**.

Thus, based on the velocity triangle between the streamwise hub velocity $U_{hub}$ and the vertical maximum velocity induced by **H**, the maximum inflow skew angle to which the porous disk is subjected during the heave motion is less than $\pm 1°$.

Following the wake deflection model of Bastankhah and Porté-Agel (2016), a 1° inflow skew angle induces a vertical wake displacement of about $0.01D$ at $x = 8.125D$, while the actual one is $0.1D$. This is consistent with the wind tunnel measurements of Bastankhah and Porté-Agel (2015) and Howland et al. (2016), which show that low yaw angles ($< 10°$) induce negligible far-wake displacement. Thus, heave motion impacts on the wake at this amplitude and Strouhal number ($A_H = 0.03D$, $St_H = 0.09$) cannot be represented by a passive tracer approach: the perturbation created by the disk motion seems to be amplified in the wake.

Following the results of a turbine model under static tilt misalignment, Bossuyt et al. (2021) showed that due to the shear present in the inflow, a positive tilt angle implies a lower wake available power and a negative tilt angle a higher one. Thus, since the skew angle can be considered a tilt angle, the low $\widetilde{P_m}$ modifications observed could be associated with the heave motion.

## 4.2 Surge motion

Figure 14 presents a schematic view of the potential impact of surge motion on the far wake of a FOWT, with wake surface and available power modifications on the same order of magnitude as those found in the experiments. The hypothesis of the surge motion impact is that it does not move the wake in the $(y, z)$ plane but instead imposes a wake modulation, with an extension and a contraction of the wake surface in phase opposition to the available power variations.

The harmonic variations in $\widetilde{S_{wk}}$ and $\widetilde{P_m}$ illustrate a modulation of the wake surface and power according to the motion phase. These variations are synchronized, and the maximum $\widetilde{S_{wk}}$ is reached at the same phase as the maximum $\widetilde{P_m}$, in agreement with momentum conservation. This is similar to the pulsating wake dynamics observed by Messmer et al. (2024) on a rotating turbine model subjected to surge at higher Strouhal numbers ($St \in [0.25, 0.5]$ vs. $St_S = 0.11$ here). The coordinates of the wake center in this case show negligible modifications between the phases (Fig. 11). More-

over, Duan et al. (2022) observed the formation of periodical vortex rings, with surges with similar motion amplitudes but a higher Strouhal number ($St = 0.55$). The authors also noted that these periodical vortex rings are not visible after $6D$, and that then, the wake swings left and right regularly. In the present study at $8.125D$ downstream, this wake swinging is not visible, but the surface variations could correspond to the passage of the periodical vortex rings on the residual signatures.

Across all phases, $\widetilde{u_m}$ undergoes a variation of $0.03U_{hub}$ at the wake center (Fig. 11). As shown in Fig. 9, the porous disk motion induces a streamwise velocity variation of $0.09U_{hub}$ maximum in the near wake. Therefore, following the wake model of Bastankhah and Porté-Agel (2014), this theoretically implies a velocity difference lower than $0.01U_{hub}$ $8.125D$ downstream of the porous disk, which is significantly lower than the total variation in velocity seen across the phases. This shows that at this amplitude and Strouhal number ($A_S = 0.01D$, $St_S = 0.11$), surge motion impacts on the porous disk wake cannot be assimilated into a succession of steady states and that the perturbation caused by the disk motion seems to be amplified in the wake.

## 4.3 Pitch motion

Figure 15 shows a schematic view of the potential impact of the pitch motion on the far wake of a FOWT, with wake center, wake surface, and available power modifications on the same order of magnitude as those found in the experiments. Pitch motions induce a vertical translation of the wake synchronized with the wake surface and available power variations, leading to the hypothesis that its impact is a combination of the impacts of heave and surge, as observed in the previous sections.

In the present study, two pitch motion cases with different Strouhal numbers ($St_{P_{0.14}} = 0.14$ and $St_{P_{0.28}} = 0.28$) were analyzed. They show approximatively the same trends but with greater wake parameter variations for $P_{0.28}$ compared to those of $P_{0.14}$, which are caused by a motion frequency closer to the natural frequency of the wake instabilities. Indeed, Li et al. (2022) showed that sway and roll motions with similar amplitudes and with a Strouhal number of about $St = 0.2/0.3$ have the greatest impact on the far wake, and Schliffke et al. (2024) showed that a porous disk subjected to surge motion

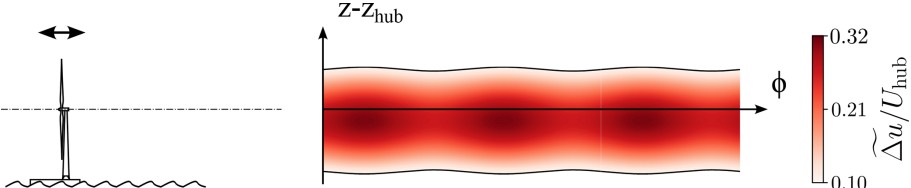

**Figure 14.** Schematic view of the surge motion impact on the wake based on the phase-averaged results. Same as Fig. 13.

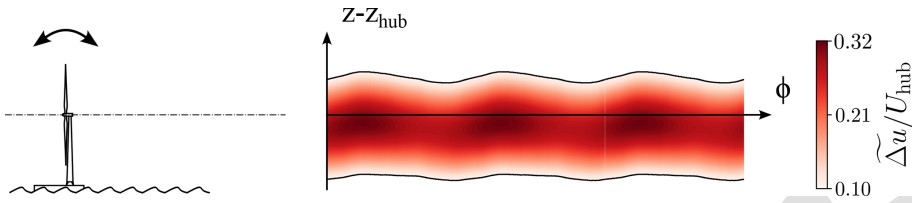

**Figure 15.** Schematic view of the pitch motion impact on the wake based on the phase-averaged results. Same as Fig. 13.

with $St = 0.35$ has a higher signature than a surge motion with $St = 0.25$ $4.6D$ downstream of the turbine model.

As seen in Fig. 4, the $4°$ pitch motions are a cinematic combination of $4°$ tilt, $0.05D$ surge, and $0.004D$ heave motions. A quasi-steady-state analysis was performed for each component of the pitch motion, divided into two categories: (i) the tilt and heave components, which induce an inflow skew deflecting the wake up or down, and (ii) the surge component, which induces wake velocity modifications. The heave component amplitude is low and induces a minimal inflow velocity modification in the ABL (Fig. 2). Thus, the velocity difference seen in the pitch motion cases is not caused by the heave component, which is why the quasi-steady-state analysis of the wake velocity modifications was only performed on the surge component.

(i) Following the wake deflection model of Bastankhah and Porté-Agel (2016), the $4°$ tilt motion induces a vertical wake displacement of about $\pm 0.06D$. On the other hand, the heave component has a motion frequency twice as high as the pitch motion, but its amplitude is too low to imply a significant inflow skew. Indeed, the velocity combination gives inflow skews of $0.2$ and $0.4°$ for $\mathbf{P_{0.14}}$ and $\mathbf{P_{0.28}}$, respectively, resulting in a vertical wake deflection lower than $0.01D$ ($0.003D$ for $\mathbf{P_{0.14}}$ and $0.005D$ for $\mathbf{P_{0.28}}$). However, across the phases, both component effects add up. With their different frequencies, the combined effects of the tilt and heave components could disturb the $z_c$ curve (Fig. 11b), resulting in a distorted sinusoid shape, as observed in the $\mathbf{P_{0.28}}$ case.

According to the results (Fig. 11), the vertical wake displacement has amplitudes of about $0.06D$ and $0.11D$, while the wake deflection of a $4°$ tilt is about $0.06D$. The phase-averaged results of the $\mathbf{P_{0.14}}$ motion case is identical to the theoretical value, while the $\mathbf{P_{0.28}}$ ones show a wake deflection higher than the theoretical one. Thus, the perturbations created by the relative angles of the inflow induced by the

pitch motion cases seem to be amplified in the far wake for $\mathbf{P_{0.28}}$ but not for $\mathbf{P_{0.14}}$.

(ii) In addition to the vertical wake displacement, the pitch motions induce a velocity difference of $0.03U_{hub}$ and $0.08U_{hub}$ at the wake center for $\mathbf{P_{0.14}}$ and $\mathbf{P_{0.28}}$, respectively. According to the wake model of Bastankhah and Porté-Agel (2014), the $0.05D$ amplitude surge component of the pitch motions theoretically implies a streamwise velocity difference lower than $0.01U_{hub}$ and slightly higher than $0.01U_{hub}$ at a downstream distance of $8.125D$. As for the surge motion case, the theoretical values are lower than the experimental ones, showing that the perturbation relative to the surge component seems to be amplified in the wake.

Further investigation is required to confirm if the porous disk under tilt-only motion exhibits these wake parameter modulations, as it will not generate an additional surge component of motion. Moreover, the heave component effect on the wake needs to be studied; as its frequency is higher, it can affect the wake even if its amplitude is relatively low.

## 5   Conclusions

The present work proposed a description of the dynamic response of a wind turbine wake observed in previous studies when a porous disk model, modeling the far wake of a turbine, is subjected to harmonic motions (Belvasi et al., 2022; Schliffke et al., 2024). All experiments were performed in the atmospheric boundary layer wind tunnel of the LHEEA at École Centrale de Nantes, where a $1:500$ neutral marine ABL was modeled. Three different platform movements are analyzed: heave ($A_H = 0.03D$, $St_H = 0.09$), surge ($A_S = 0.06D$, $St_S = 0.11$), and pitch ($A_{\mathbf{P_{0.14}}} = 4°$, $St_{\mathbf{P_{0.14}}} = 0.14$ and $A_{\mathbf{P_{0.28}}} = 4°$, $St_{\mathbf{P_{0.28}}} = 0.28$) motions.

These cases are harmonic, with realistic amplitudes and frequencies according to a full-scale $2\,\mathrm{MW}$ wind turbine combined with a barge-type floater. The chosen ampli-

tudes/frequencies are relative to the second-order motions due to the response of the floater linked to mooring lines and anchoring characteristics. A stereo-PIV system was used, measuring the three-component velocity field in a plane normal to the freestream flow $8.125D$ downstream of the model. Phase-averaging with kernel smoothing was applied to the velocity fields represented in the fixed and the moving frames of reference in order to study the motion impacts on the wake and their effects on a potential downwind turbine. The results suggest that the floater movements add coherent spatiotemporal behaviors to the wake of a FOWT by modulating the crosswise wake positions, the wake surface, and the available power, with amplitudes higher than those expected using basic quasi-steady-state approaches – using the wake model of Bastankhah and Porté-Agel (2014) and the wake deflection model of Bastankhah and Porté-Agel (2016). Thus, several hypotheses and conclusions have been drawn.

– The moving frame of reference calculation method and the phase-averaging with kernel smoothing algorithm used in this study enable the observation of coherent spatiotemporal wake behavior of a turbine model under realistic turbulence conditions. Thus, the first method avoids the wake parameter misestimations caused by the presence of wake meandering due to the large turbulent structures present in the inflow, and the second method separates the periodic velocity fluctuations due to motion and the background turbulence.

– Heave motion translates the wake vertically, with an amplitude higher than the motion itself. The inconsistent evolution of the wake surface and the available power variations might be partly associated with a processing bias and a ground effect. Moreover, this can be due to the shear present in the inflow seen in previous static tilt misaligned turbine studies – e.g., Bossuyt et al. (2021). Indeed, the skew angle created by the heave motion could deflect the wake, similar to a tilt misalignment.

– Surge motion leads to contraction and expansion of the wake surface in the crosswise plane, with negligible wake displacement, and modifies the available power within the wake. The results show that wake crosswise surface and velocity modulations are in phase opposition: a large wake surface implies low power in the wake and vice versa, consistent with momentum conservation.

– Pitch motion involves a combination of the heave and surge motions. For the heave motion, the wake is translated vertically, and, for the surge motion, the wake surface and available power values are modulated in phase. The results show that the two wake dynamics are synchronized: when the wake goes to its highest point, it has a large surface area and low available power. However, this synchronization is phase shifted by $\frac{\pi}{3}$, partly caused by the processing bias and by a ground effect in the same way as in the heave motion case, or it is caused by the heave component, which is intrinsic to pitch motion that has twice its frequency.

These observations present similarities with previous work on either heave (Kleine et al., 2022; Li et al., 2022), surge (Duan et al., 2022; Messmer et al., 2024), or pitch (Kleine et al., 2022) motions. The Strouhal numbers of the motions investigated are lower than those of maximal impact found in previous studies (Li et al., 2022; Messmer et al., 2024); one expects that applying similar Strouhal numbers to our configuration would provide higher wake parameter modifications.

Despite the use of a porous disk that prevents the near-wake flow from the presence of tip vortices and rotational momentum, the wake dynamics are modified by the disk motions. It is assumed that the present wake dynamics are directly the result of flow perturbations initiated by the porous disk motions in the near wake and amplified in the far wake. This diverges from previous interpretations arguing that the flow perturbations related to floating movements are due to the tip vortices that are impacted by the turbine motion and interacting with each other (Kleine et al., 2022). A comparison between a porous disk and a rotating model, with realistic power and thrust curves, both immersed in an atmospheric boundary layer and subjected to floating motions, is needed to distinguish between the effective influence of the tip vortices and/or rotational momentum on wake dynamics.

The large majority of previous studies analyzed the impact of the floating motions on the wake of a turbine using harmonic motions, while the full-scale ones are present in a range of amplitudes and frequencies. This concentration of the motion energy into a single frequency present limitations; Schliffke et al. (2024) observed that with the same amplitudes, the energy associated with harmonic motions is higher than that of multifrequency motions. Thus, further investigation with realistic motions – i.e., with a range of amplitudes and frequencies rather than harmonic motion – is necessary to observe the actual impact of the floating motions.

**Data availability.** The raw data for each final figure are available in a Zenodo dataset (https://doi.org/10.5281/zenodo.15038753; Aubrun and Conan, 2025).

**Supplement.** The supplement related to this article is available online at [the link will be implemented upon publication].

**Author contributions.** BC designed and performed the experiments. AH coded the postprocessing methodology and processed and analyzed the data under the supervision of and in discussion with BC and SA. SA was responsible for funding acquisition and project administration. The original draft was written by AH and was reviewed and edited by SA and BC.

**Competing interests.** At least one of the (co-)authors is a member of the editorial board of *Wind Energy Science*. The peer-review process was guided by an independent editor, and the authors also have no other competing interests to declare.

**Disclaimer.** Publisher's note: Copernicus Publications remains neutral with regard to jurisdictional claims made in the text, published maps, institutional affiliations, or any other geographical representation in this paper. While Copernicus Publications makes every effort to include appropriate place names, the final responsibility lies with the authors.

**Acknowledgements.** The authors wish to acknowledge Titouan Olivier-Martin for helping with the installation of the wind tunnel experimental setup. Additionally, credit is due to Thibaud Piquet for his impressive support in getting the S-PIV system operational despite the many twists and turns.

**Financial support.** This research has been supported by the West Atlantic Marine Energy Community (WEAMEC), the Pays de la Loire Region, and the École Centrale de Nantes.

**Review statement.** This paper was edited by Raúl Bayoán Cal and reviewed by two anonymous referees.

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

## Remarks from the language copy-editor

CE1     Should this affiliation then be "École Centrale Nantes, Nantes Université"? We otherwise require departments for university affiliations.

CE2     Please note the changes throughout in line with our standards for ranges.

CE3     The degree signs were in the original accepted manuscript. A change to a value such as this would require approval from the handling editor. If you would like us to start this process, please provide a statement detailing why this change must be implemented.

CE4     Please note the change to commas in this expression throughout.

## Remarks from the typesetter

TS1     Since $H$, $S$, $P_{0.14}$ and $P_{0.28}$ seem to be variables, can they be made italics throughout?

TS2     Please provide date of last access.

TS3     Please provide date of last access.

TS4     Please provide date of last access.

TS5     Please provide date of last access.

TS6     Please provide date of last access.