# Peer review of "Spatio-temporal behavior of the far-wake of a wind turbine model subjected to harmonic motions: Phase averaging applied to Stereo-PIV measurements"

_Wind Energy Science, 2024_

## Referee Comment (RC1)

*Spatio-temporal ehavior of the far-wake of a wind Turbine model subjected to harmonic motions: Phase averaging applied to Stereo-PIV measurement.*

*authors:*
Antonin Huberta, Boris Conana, and Sandrine Aubrun

**Summary:**

The manuscript entitled "Spatio-temporal behavior of the far-wake of a wind Turbine model subjected to harmonic motions: Phase averaging applied to Stereo-PIV measurement" endeavors to describe periodic influences in the advection of the wake of a floating offshore wind turbine that arise from heave, surge, and pitch motions. The methods employed by the authors are well-founded and build on a rich history of wind tunnel research with a set of porous discs that are now familiar in the literature. While the authors austensibly focus on the phenomena of wake meandering, very little effort is made to connect the resultant phase-averaged wake trajectories to the underlying mechanisms driving wake meandering. The work would be made more impactful overall by connecting the results with model that is use widely in the wind energy engineering space, such as the dynamic wake meandering model or a wake-added turbulence model. As an alternative it would be nice to see the authors connect the observed wake behaviors, such as the large period of vertical wake meandering relative to the heave motion, with broader constraints such as exchanges between the ABL and offshore wind plants.

**Comments:**

- In the description of the experiment, I'm left wondering how representative the modeled boundary layer is to the real marine ABL that will be seen by operating FOWTs. The authors make a passing comparison to conditions described by ESDU (1985), but it's not clear how similar these conditions are to offshore development areas around the world. Readers of this research would be more able to integrate these findings into their own work if it were more clear what the target conditions are, what region they represent, etc. Please contextualize the boundary layer profiles and boundary conditions (roughness, shear exponent, etc.) with respect to actual obesrved quantities.
- The authors do not justify why the SPIV measurements focus on a single transverse plane $8.125D$ downstream of the modeled turbine. This location is relatively far in to the wake. At this distance, we expect the wake to break up in many cases, complicating the identification of closed velocity contours and regular periodic motion. We should also expect trajectories to depend on the downstream coordinate, such as a net vertical displacement of the wake, that cannot be described completely with measurements at a single locations.
- In Table 1. the motion of the full-scale turbine is decribed in terms of amplitude and meandering period. For the model-scale turbine, the motion is described in amplitude and frequency. Why present them differently? It is also not clear what the authors mean by "normalized amplitude."

Normalized by what? How representative are the Strouhal numbers of the modeled scale vs the full scale? I presume that the platform motion for the FOWT are driven at specific Strouhal numbers, rather than arising from hydrodynamic forcing, but this isn't explicitly stated in the paper.

- Line 167 — $\sigma$ should have units of length.
- Equation 5 and throughout — multiplication is implied with a period, but should probably use the `\cdot` macro.
- Figure 6 — it would be far more interesting to plot the estimated wake center data, rather than the sinusoidal with noise. This would help the reader understand what the actual data look like, and how the period behavior is quantified.
- Figure 7 and throughout — some of the vertical axis labels are not rendered correctly and are missing subscripts.
- Figure 11 — It would be much easier to understand these results with error bars or uncertainty estimates in the trends. Also, the authors should comment on the complexity evident in the P0.28 case. Is there some non-linearity or more than a single frequency relevant to the wake center trajectory? As a more general question, how are the authors confident that a simple sinusoidal relationship is sufficient to capture the complexity of the modulation in the wake?
- Figure 12 — the phase-averaged surface metric for the heave case does not match conceptual diagram in Fig. 13. I would expect the surface of the wake to be approximately constant in time, since the authors suggest that the main change is periodic vertical displacement. At the very least, the results and discussion suggest that the wake surface for the heave case should change less than for the surge case, which should show period contraction and expansion.
- The only model mentioned in the manuscript is the Jiménez model from 2010, which described lateral or vertical wake deflection due to static yaw offsets. Without framing the results of this study in terms of a model or underlying physical relationship that can be used to explain the observations, this work will have very limited impact in the field of offshore wind energy.
- Line 435 — the authors state that "Heave motion translates the wake vertically with an amplitude higher than the motion itself." This observation likely arises from the fact that the wake is expanding as it evolves downstream and interacts with turbulence in the inflow boundary layer. If this observation is stating that there is some mechanism amplifying vertical wake motion, it could have pretty big implications for energy fluxes and exchanges between wind turbines and the ABL. Please elaborate.
- line 438 — The authors claim that, "Surge motion leads to contraction and expansion of the wake surface in the crosswise plane, with negligible wake displacement, ..." Is this insight supported by the results in figures 11 and 12? Is the wake center moving vertically or laterally for the surge case?

---

## Author Comment (AC1)

*Spatio-temporal behavior of the far-wake of a wind Turbine model subjected to harmonic motions: Phase averaging applied to Stereo-PIV measurement.*

*authors:*
Antonin Hubert, Boris Conan, and Sandrine Aubrun

The authors would like to express their sincere thanks to both reviewers for their careful assessment. This document responds to each of the points raised. Reviewers remarks are in black, authors answers in blue. A "diff" file is also provided to enlighten the modifications made to the manuscript.

The line numbers given in the authors' answers correspond to the ones of this file.

**Summary:**

The manuscript entitled "Spatio-temporal behavior of the far-wake of a wind Turbine model subjected to harmonic motions: Phase averaging applied to Stereo-PIV measurement" endeavors to describe periodic influences in the advection of the wake of a floating offshore wind turbine that arise from heave, surge, and pitch motions. The methods employed by the authors are well-founded and build on a rich history of wind tunnel research with a set of porous discs that are now familiar in the literature. While the authors austensibly focus on the phenomena of wake meandering, very little effort is made to connect the resultant phase-averaged wake trajectories to the underlying mechanisms driving wake meandering. The work would be made more impactful overall by connecting the results with model that is used widely in the wind energy engineering space, such as the dynamic wake meandering model or a wake-added turbulence model. As an alternative it would be nice to see the authors connect the observed wake behaviors, such as the large period of vertical wake meandering relative to the heave motion, with broader constraints such as exchanges between the ABL and offshore wind plants.

The focus of this article is the analysis of the wake dynamics in response to floating motions, not on wake meandering due to external forcing from- the large-scales of turbulence present in the inflow, even if both are observed in the present study thanks to realistic atmospheric conditions reproduced in the wind tunnel. We observed vertical harmonic wake meandering in the far-wake of the disc subjected to heave and pitch motions, but it is related to the motion itself. Thus, the underlying mechanisms driving the wake meandering due to the large-scales of turbulence are not relevant to explain the phenomena observed in this article. To avoid any confusion, a paragraph has been added to explain these two sources of meandering.

Added to the article line 96: "This phenomenon should not be confused with the motion-induced wake meandering observed in previously cited studies; wake meandering signifies a displacement of the global wake in a crosswise direction, but this can be caused by the turbulent large-scale structures present in the inflow - thus appearing in the wake of both bottom-fixed and floating wind turbines - or by the motion of the floating platform - thus only appearing in the wake of FOWT".

In this article, comparisons are performed with static wake models used in the wind energy engineering community (Jensen static wake model and wake deflection model of Jimenez). Such models are used in FLORIS, and have already been validated (Doekemeijer et al. 2022). However, more precise models are used in the revised version; the wake model of Bastankhah and Porté-Agel (2014), and the wake deflection model of Bastankhah (2016). Comparisons with dynamic wake meandering and wake-added

turbulence models are outlooks that could be achieved in another paper. Here, the authors wanted to focus on the observation of phenomena related to floating motion under realistic conditions, by using phase-averaged and kernel smoothing algorithm, something never done before.

**Comments:**

- In the description of the experiment, I'm left wondering how representative the modeled boundary layer is to the real marine ABL that will be seen by operating FOWTs. The authors make a passing comparison to conditions described by ESDU (1985), but it's not clear how similar these conditions are to offshore development areas around the world. Readers of this research would be more able to integrate these findings into their own work if it were more clear what the target conditions are, what region they represent, etc. Please contextualize the boundary layer profiles and boundary conditions (roughness, shear exponent, etc.) with respect to actual observed quantities.

  These models have already been validated by full-scale experimentation results, and are largely accepted in the atmospheric boundary layer community (Kaimal and Finnigan, 1994). Moreover, in this article the authors limited their research to the neutral conditions of thermal stability of the ABL, which are idealised conditions of actual observed quantities.

  Added to the article line 153: "The ABL parametrisation and its dependence to the type of terrain have been largely validated through observational statistics (Kaimal and Finnigan, 1994, Counihan, 1975) and led to guidelines on the physical modelling of such ABL in wind tunnel (VDI, 2000). Nevertheless, the potential modification of the marine ABL according to the sea state is disregarded in the present study; the complexity of the wind-wave-wake interactions are not fully modelled, and can impact the observed results (Porchetta et al. 2019; 2021; Fercak et al. 2022)."

- The authors do not justify why the SPIV measurements focus on a single transverse plane 8.125D downstream of the modeled turbine. This location is relatively far in to the wake. At this distance, we expect the wake to break up in many cases, complicating the identification of closed velocity contours and regular periodic motion. We should also expect trajectories to depend on the downstream coordinate, such as a net vertical displacement of the wake, that cannot be described completely with measurements at a single locations.

  This distance of 8.125D is realistic compared to full-scale distances between two wind turbines in a wind farm (Commission et al. 2018). This distance corresponds also to previous experimentations performed in similar conditions (Schliffke et al. 2022, Belvasi et al. 2022, Schliffke et al. 2024).

  Added to the article line 185: "This value corresponds to the previous experimentations done by Schliffke et al. (2022); Belvasi et al. (2022); Schliffke et al. (2024) to observe FOWT wake dynamics. Moreover, this 8.125D value is realistic compared to full-scale distances between two wind turbines in a wind farm (Commission 2018)."

- In Table 1. the motion of the full-scale turbine is decribed in terms of amplitude and meandering period. For the model-scale turbine, the motion is described in amplitude and frequency. Why present them differently?

These values are motion parameters (amplitudes and periods/frequencies of the motions, either in full or reduced scales), not meandering amplitude/period of the wake. In full-scale, the ocean engineering community commonly uses the amplitude and period values to describe a floating motion, more suitable since the values are large. On the other hand, in the wind energy community, the motions are described in amplitude and frequency. Thus, they are presented differently.

It is also not clear what the authors mean by "normalized amplitude." Normalized by what?

The amplitudes are normalised by D. Added to the article line 180: "Amplitudes are normalised by D".

How representative are the Strouhal numbers of the modeled scale vs the full scale? I presume that the platform motion for the FOWT are driven at specific Strouhal numbers, rather than arising from hydrodynamic forcing, but this isn't explicitly stated in the paper.

Line 179, it is noted in the article: "Full scale configurations were downscaled to wind tunnel configurations by conserving the same normalised amplitudes and Strouhal numbers of the motions". The motions investigated in this article are representative of realistic floating motions, it is noted in the article line 170: "The motion amplitudes and frequencies of a barge-type platform were extracted from a data base of numerical simulations provided by BW-Ideol, and are specific to low-frequency motions related to the mooring lines acting on the floating platform". Modifications line 171: "… specific to the second-order motions related to the mooring lines and anchors acting on the floating platform".

- Line 167 – $\sigma$ should have units of length.

  Modification in line 210: "$\sigma = 0.26D$"

- Equation 5 and throughout – multiplication is implied with a period, but should probably use the \cdot macro.

  Modification made in Eq 1, 2, 3, 4, 5, 6, 8

- Figure 6 – it would be far more interesting to plot the estimated wake center data, rather than the sinusoidal with noise. This would help the reader understand what the actual data look like, and how the period behavior is quantified.

  The only values computed with the phase-averaging and kernel smoothing algorithm are the velocity deficit fields in FFoR and MFoR; the estimated phase-averaged wake centre coordinates are directly calculated with a Gaussian fitting using the phase-averaged velocity deficit fields in FFoR, as explained in line 234: "the WGC method is applied to the instantaneous velocity fields, and Gaussian fitting to the phase-averaged ones", and in Fig. 8. Thus, perform the same figure with the estimated wake centre coordinates would not be coherent, and it would be confusing with the velocity deficit fields.

- Figure 7 and throughout – some of the vertical axis labels are not rendered correctly and are missing subscripts.

  This problem neither appears in the primary pdf nor the downloaded one for the authors. Might it be caused by an issue with the pdf reader software?

- Figure 11 – It would be much easier to understand these results with error bars or uncertainty estimates in the trends.

Statistical uncertainties are already present, but too small to be visible in this figure. It is noted line 349: "The coloured zones, representing the statistical uncertainties defined in Eq. 7, are not visible due to their small values."

Also, the authors should comment on the complexity evident in the P0.28 case. Is there some non-linearity or more than a single frequency relevant to the wake center trajectory?

The complexity of pitch motion is detailed in the methodology section and displayed on Fig. 4. It is noted line 180: "The pitch motion has a rotation centre located at the floater level, and can be considered as a combination of tilt (pitch with a rotation axis at the disc centre), surge and heave motions: the 4° amplitude corresponds to a 8.4 mm amplitude surge with a 0.3 mm amplitude heave, as visible in Fig. 4".

Moreover, the complexity observed with the curve is discussed in Sect. 4.3, line 451: "With their different frequencies, the combined effects of the tilt and heave components could disturb the zc curve (Fig. 11 (b)), resulting in a distorted sinusoid shape". Added in line 453: "... as observed in P0.28 case".

As a more general question, how are the authors confident that a simple sinusoidal relationship is sufficient to capture the complexity of the modulation in the wake?

Previous studies show that the spectra of the wake parameters of a turbine model subjected to harmonic motion show one clear frequency signature appearing at the exact motion frequency (Bayati et al. 2017, Fu et al. 2019, Belvasi et al. 2022, Schliffke et al. 2024). This is explained in the introduction, line 120: "they observed clear signatures of the harmonic motion frequencies in the spectra of the wake parameters". Modification line 121: "they observed clear unique signatures...". Thus, the impact of the motion – i.e. what is studied in this article - is mainly present at a single frequency.

This is a limitation since, in the full-scale, floating platforms move at a range of -frequencies rather than a single one. However, this is a limitation largely accepted in the FOWT community, the large majority of previous studies are performed with harmonic motions.

Added to the line 517: "The large majority of previous studies analysed the impact of the floating motions on the wake of a turbine using harmonic motions, while the full-scale ones are present in a range of amplitudes and frequencies. This concentration of the motion energy into a single frequency present limitations; Schliffke et al. (2024) observed that, with same amplitudes, the energy associated to harmonic motions is higher than that of multi-frequency motions. Thus, further investigations with realistic motions - i.e. ones with a range of amplitudes and frequencies rather than harmonic ones - are necessary to observe the actual impact of the floating motions."

- Figure 12 – the phase-averaged surface metric for the heave case does not match conceptual diagram in Fig. 13. I would expect the surface of the wake to be approximately constant in time, since the authors suggest that the main change is periodic vertical displacement. At the very least, the results and discussion suggest that the wake surface for the heave case should change less than for the surge case, which should show period contraction and expansion.

The wake surface modifications for the heave case are mainly explained with the algorithm bias, described in the article line 363: "as mentioned earlier in Sect. 2.3, the truncated part of the wake may lead to an overestimation of the actual wake surface variations. Indeed, the cases showing the greatest Swk modifications are those with the highest zc ones". Thus, the

authors did not want to be confusing, and remove the phase-averaged wake surface modulation in this figure. However, the figure have been changed to correspond to those values.

The only model mentioned in the manuscript is the Jiménez model from 2010, which described lateral or vertical wake deflection due to static yaw offsets. Without framing the results of this study in terms of a model or underlying physical relationship that can be used to explain the observations, this work will have very limited impact in the field of offshore wind energy.

Cf. response in the summary (3rd paragraph).

- Line 435 – the authors state that "Heave motion translates the wake vertically with an amplitude higher than the motion itself." This observation likely arises from the fact that the wake is expanding as it evolves downstream and interacts with turbulence in the inflow boundary layer. If this observation is stating that there is some mechanism amplifying vertical wake motion, it could have pretty big implications for energy fluxes and exchanges between wind turbines and the ABL. Please elaborate.

The purpose of this article is to observe wake dynamics, and a future article might try to elaborate a FOWT wake model based on these observations. Here, quasi-steady-state analysis are performed with the wake models, showing that the perturbations created by the floating motions are amplified in the far-wake, for all motions not only for heave. It is noted lines 402, 428, 461, and 468. It is also noted in the conclusion line 483: "The results suggest that the floater movements add coherent spatio-temporal behaviours to the wake of a FOWT, by modulating the cross-wise wake positions, the wake surface, and the available power, with amplitudes higher than those expected by using basic quasi-steady-state approaches".

- line 438 – The authors claim that, "Surge motion leads to contraction and expansion of the wake surface in the crosswise plane, with negligible wake displacement, ..." Is this insight supported by the results in figures 11 and 12? Is the wake center moving vertically or laterally for the surge case?

The results show that the wake, in surge motion, is not moved in the crosswise plan (visible in Fig. 11). Added to the article line 416: "The coordinates of the wake centre, in this case, show negligible modifications along the phases (Fig. 11)".

The line numbers given in the authors' answers correspond to the ones of this file.

In this paper PIV measurements are used to study the dynamic behavior of the wake of an oscillating porous disk. In this way the study aims to gain more insights in wake dynamics of offshore floating wind turbines. The authors discuss the used approximations of their method and its limitations in the text (i.e. the porous disk approximation). However, it would also be good to mention the Reynolds number, and the fact that this study does not include the interaction with the dynamic ocean surface, which is also important for floating turbines and the spatio-temporal wake development. Instead of time-averaged wake analysis, the authors use conditional averaging of the wake velocity to capture its periodic behavior. Wake analysis is performed using a classical fixed frame of reference, and a moving frame of reference in which the meandering of the wake center in the measurement plane is followed.

In this article, the Reynolds number is approximately Re = 3x10^4. A Reynolds number independence study has been performed in a previous article (Schliffke et al. 2022), and small deviations are observed in velocity, turbulence, and TKE profiles, but still the assumption of Reynolds number independence is valid.

Added to the article line 159: "(Reynolds number of $Re=3.10^4$)".

Added to the article line 161: "Moreover, this same study showed that the assumption of Reynolds number independence is valid".

Indeed, the interaction with the dynamic ocean surface is not represented.

Added to the article line 155: "Nevertheless, the potential modification of the marine ABL according to the sea state is disregarded in the present study; the complexity of the wind-wave-wake interactions are not fully modelled, and can impact the observed results (Porchetta et al. 2019; 2021; Fercak et al. 2022)".

The results and discussion in the paper are a valuable contribution to the study of wakes of floating wind turbines. The paper is generally very well written, with clear figures, and analyses. Below several minor comment for the authors:

- line 4': Previous studies showed that harmonic motions with realistic amplitude and frequency and under a modelled atmospheric boundary layer have no significant impact on time-averaged values, but that frequency signatures are still visible in spectra of wake parameters.' —> The reviewer finds this a confusing statement, and an incorrect generalization:
    - By definition the spectral content has an impact on the time averaged statistics. It is however indeed possible that in a turbulent boundary layer or flow with high background turbulence levels the impact on the time averaged statistics becomes masked or is relatively small, but there should be some connection to the statistics.

    Indeed, in the mentioned articles here have high background turbulence intensity (Belvasi et al. 2022, Schliffke et al. 2024, Li et al. 2024). Modification in line 5: "no significant impact on time-averaged values due to the relative high background turbulence, ..."

    - This conclusion depends strongly on the measured conditions (—background turbulence, motion frequency and amplitude of the turbine), and the measured location (near wake vs far wake). There are studies that show a meaningful difference in the time averaged mean wake properties, showing that wake recovery and spreading can be affected, and others where it is indeed small. But be careful in generalizing this statement.

The studies showing this statement are mentioned; here the authors are interested in motions with precise combination of amplitude and frequency, under high turbulent inlet conditions. Modification in line 4: "Some previous study…".

- line 98: similar comment: Be careful with this conclusion, this is true if there is a significant amount of background turbulence, for the tested conditions, and for the tested motion amplitude and frequencies, as stated in the sentence above. However, there are also studies that have seen an impact on time averaged statistics. Possibly because of the different conditions, motions, or the use of a rotor model? As an extreme example: a very slow motion / at a very small Strouhal frequency, will visibly spread the wake out, and thus affect time-averaged as well as spectral properties.

  Indeed, this is very dependent of the experimental conditions, which are very specific in this article and specified in line 118: "measurements with a porous disc subjected to low Strouhal number heave, surge and pitch motions", and line 123: "they showed that, because of the high level of turbulence, the shear  layer and the presence of meandering, due to the ABL modelling, the  conventional time-averaged results are inappropriate".

  Added to the article line 126: "in similar motion conditions".

- line 120: It can be interesting to add the length over which the boundary layer is developed in the wind tunnel.

  Added to the article line 148: ", and developed over a total length of 20 m".

- line 134: It is best to note that this is a 'hypothetical', 'fictive' or 'representative' power coefficient, given that it is a porous disk.

  Added to the article line 167: "a representative…".

- line 150: Can the authors provide more information about the laser sheet thickness needed for the measurements, and the estimated measurement uncertainty of the PIV velocities?

  Added to the article line 189: "and a thickness of approximately 3 mm…".

  The measurement uncertainty of the SPIV system is difficult to assess. However, an estimation can be given.

  Added to the article line 191: "The velocity measurement uncertainty of SPIV systems is a combination of the numerous uncertainties present in the measurement chain, and is related to the installation and to the post-processing algorithms (Raffel et al. 1998, Wieneke 2017, Sciacchitano 2019). Adrian et al. (2011) stated that a typical value of the SPIV measurement uncertainty displacement of the particles is 0.1 pixel units. However, this is highly simplistic and should be treated with caution since, as mentioned earlier, the uncertainties vary with the experimental set-up".

- line 195: The PIV measurements are performed pretty far downstream (x/D=8.125), where the wake is likely overwhelmed by ambient turbulence from the turbulent boundary layer. This must make it challenging to pinpoint the wake center in instantaneous snapshots given the broken up wake shape.  Given that the instantaneous wake shapes are likely very irregular / dispersed at this distance, can the larger uncertainty of the wake center affect the analysis in this paper in any way? For example: can errors of wake center add artificial 'meandering' to the analysis of the MFoR?

Indeed, the uncertainty can affect the analysis, especially in MFoR. However, the statistical uncertainties of the phase-averaged wake parameters are calculated with those of the phase-averaged velocity deficit fields in MFoR.

Moreover, this is why the WGC algorithm is preferred here, in the context of a high turbulent intensity inflow, instead of other wake centre estimation algorithms, which are used in other studies, such as the convolution or the Gaussian fitting. Thus, the present authors performed a study that compares the different wake centre estimation algorithms found in the literature and showed that, in contrary to the others, WGC one was the most robust in flow configurations similar to the present one, permitting to estimate wake centre even with high turbulence (Hubert et al. 2022).

- line 217: Do the authors find similar conclusions if a different value is used? How sensitive are the conclusions to this value?

This is a good remark, the chosen value is very subjective. Moreover, this value changes according to the chosen kernel. As the kernel smoother acts as a filter, where $\lambda$ would correspond to the filter window size; the higher $\lambda$, the smoother the curves. Thus, the value of $\lambda$ is a compromise between minimising fluctuations of the curves without flattening too much the amplitudes of the phenomena being studied. This is explained line 255: "Nevertheless, the resulting values of kernel smoothing must be taken with caution, as the method acts like a low-pass filter and tends to limit extreme phenomena. Also, if a kernel function is too narrow, the result is based on too few data and gives too much weight to each particular piece of data, resulting in an under-smoothed estimation. Conversely, if the kernel function is too large, the result takes too many data into account, resulting in an over-smoothed estimation. Thus, the choice of the bandwidth value $\lambda$ is important".

This is also why the results are preferably analysed through the curve trends rather than actual values. However, previous iterative tests showed that $\lambda$ needs to be significantly changed to impact the phase-averaged results.

- line 235: One has to be very careful with this sentence. For a turbine at a fixed downstream location the FFoR is still what matters in determining the available downstream power. In that case it is generally not relevant for the downstream turbine if the wake power is lower or higher in a MFoR, unless when temporal interaction like dynamic loading are investigated. The reviewer agrees however that it can be interesting from a wake modeling point of view to separate the impact of wake meandering in the MFoR approach. On the other hand, for a floating turbine with variable position, both the MFoR and the FFoR are not a complete description because the spatio-temporal characteristics of the wake need to be considered in combination with the dynamic motion of the downstream turbine.

In this article, the FFoR approach serves as the study of the impact of the turbine motions on a potential downstream turbine, while MFoR serves as the study of the wake itself.

Added in line 264: "in the Fixed Frame of Reference (FFoR), to observe the impact of the turbine motion on a potential downstream turbine, and in the Moving Frame of Reference (MFoR), to study the wake under floating motions.".

Indeed, considering FOWT, the downstream turbine will move, and MFoR and FFoR are not a complete description. Thus, in this article, the downstream turbine is assumed to be fixed, and

the observed wake parameter modulations are taken as estimations and trends rather than actual values.

- line 306: Figure 11 is however also an interesting graph because it shows that pitch is not just surge + heave, due to the angular misalignment of the rotor. Due to the pitch angle the porous disk deflects the wake up or down, generating a lift force, accompanied by a counter rotating vortex pair. An interaction of this CVP with the shear in the boundary layer, and possibly the presence of a tower could explain why pitch affects the y-location slightly, and periodically.

  Indeed, the vertical wake deflection is explained in the discussion section 4.3 line 446: "Following the wake deflection model of Jiménez et al. (2010), the 4° tilt motion induces a vertical wake displacement of about 0.08D" (now with the wake deflection model of Bastankhak and Porté-Agel (2016). No investigation has been performed on the possible signature of CVP in the crosswise velocity fields, but the authors note this interesting remark for a future study.

- line 389 : It would be helpful to add to this sentence that there is also a geometric misalignment angle for pitch, deflecting the wake up or down, on top of the heave and surge motion.

  This is mentioned through the sentence line 441: "the tilt and heave components, which induce an inflow skew".

  Added to the line 442: "... deflecting the wake up or down,".

- line 417: Using the word 'turbine model' would insinuate that the tests were done with an actual rotor. 'porous disk model' is the most correct wording.

  Modifications are performed line 474: "porous disc model, modelling the far-wake of a wind turbine".

- line 423: From the sentence it is not clear what is meant: the second-order motions should be relatively smaller than the simulated amplitudes/frequencies? Or what is meant with 'second-order' ? larger/smaller / a background motion on which the pitch/heave/etc are superimposed?

  The second-order motions are related to the second frequency peak observed in the spectra of the floating platform motions provided by BW-Ideol. The first-order motions are associated to the response of the floater to wave solicitations, while the second-order ones are associated to the response of the floater interacting with the mooring lines and anchoring systems. This is explained in the previous study performed by Schliffke et al. (2024).

  Added to the line 171: "- the first order being related to the response of the floater to wave-to-wave solicitations (Schliffke et al. (2024)"

- line 428: This is a main conclusion for the paper, and it is also in agreement with results in the literature. Can the authors discuss or comment on the agreement with results in the literature?

  Indeed, the authors added to the article a sentence concerning the similarities with previous studies line 508 (Cf. correction for line 454).

- line 432: As discussed in a previous comment: In my opinion this is only a misrepresentation if one wants to understand the characteristics of instantaneous wake properties without the impact of meandering. For the purpose of characterizing the available power for a fixed downstream turbine, there is no misrepresentation with the FFoR method.

In fact, the authors aimed to use the MFoR to better characterise the wake-only properties (freeing itself from the meandering) of the wind turbine. This is performed with the characterisation of the wake surface, and the velocity within the wake (evaluated by the available power in MFoR), and in FFoR these values are mis-estimated due to the wake movement. However, the available power in FFoR is used in the context of the study of the impact of the turbine motions on a potential downstream turbine, and is not mis-estimated in this case.

- line 438-439: this conclusion seems to be in agreement with findings in the literature. In that case, it would be helpful to discuss the agreement.

Indeed, the authors added to the article a sentence concerning the similarities with previous studies line 508 (Cf. correction for line 454).

- line 439: 'The results show that.. ' This sentence is not clear. If there is momentum conservation the wake would have the same power independent of the area? Or is the variation in power in the wake a result of the porous disk creating a stronger wake when it moves forward (higher velocity difference), and a smaller wake when it moves backwards, as also modeled by the authors using the wake model? Can the authors elaborate more clearly?

After some calculations (not shown in the article), the power within the wake is still modified by the motion. However; this could be caused by the S-PIV measurement plane which truncates the wake at some points. Nevertheless, a smaller wake surface implies a higher velocity within the wake, and a higher wake surface implies a lower velocity within the wake. Thus, in accordance with the momentum conservation.

Moreover, the relative position of the porous disc with the variation of power cannot be concluded by the actual results, as the time between the perturbation created by the porous disc motion in its near-wake and its visualisation in the far-wake is not known. A future work on the synchronisation between motion and phase-averaged results should be done.

- line 444: 'when the wake goes to its highest point, it has a large surface and a low available power' Are any of these observations in agreement with what is available in the literature? For example, studies of static tilt misaligned turbine models also find higher wake deficit when the wake is deflected upwards.

Indeed, the literature about tilt misaligned turbine also finds a higher velocity deficit when the wake is deflected upwards and inversely for downward deflection. This is due to the shear present in the inflow. This is consistent with the findings of the heave motion case, creating a skew angle (relative angle between the disc and the inflow).

Added to the article line 404: "Following the results of a turbine model under static tilt misalignment, Bossuyt et al. (2021) showed that, due to the shear present in the inflow, a positive tilt angle implies a lower wake available power and a negative tilt angle, a higher one. Thus, since the skew angle can be considered as a tilt angle, the low Pm modifications observed could be associated to the heave motion."

Added to the article line 495: "Moreover, this can be due to the shear present in the inflow seen in previous static tilt misaligned turbine studies – e.g. (Bossuyt et al. 2021). Indeed, the skew angle created by the heave motion could deflect the wake, similar to a tilt misalignment."

- line 454: There are in fact experiments in the literature with rotating turbine models subject to floating motions, some also with conditionally averaged wake analyses. Do the authors find agreements between their findings which can also be used to strengthen their results and porous disk approach?

  Some observations are similar. However, this outlook says that a complete study between a porous disc and a rotating model has to be performed to see if the model has an impact on the results. The results showing similar observations are added to the conclusion.

  Added to the article line 508: "These observations present similarities with previous works, either for motions of heave (Kleine et al. 2022; Li et al. 2022), surge (Duan et al. 2022; Messmer et al. 2024), or pitch (Kleine et al. 2022)".

  The observations of these studies are mentioned in the introduction of the article lines 56 (Kleine et al. 2022), 47 (Li et al. 2022), 59 (Duan et al. 2022), and 63 (Messmer et al. 2024).

[revised manuscript text omitted]

---

## Author Response (AR2)

The authors would like to express, again, their sincere thanks to both reviewers for their careful assessment. This document responds to each of the points raised. Reviewers remarks are in black, authors answers in blue. A "diff" file is also provided to enlighten the modifications made to the manuscript.

The line numbers given in the authors' answers correspond to the ones of this file.

A very interesting and well written paper, a few small comments & thoughts:

• line 111: 'focusing the problem on the wake instabilities'. It's a detail, but it would be more clear if the authors make it 'far-wake instabilities'.

Modifications in line 15: "far-wake instabilities".

• line 494: To provide context for the conclusions, it would be helpful to summarize the tested Strouhal frequencies and normalized amplitudes in the conclusion section.

Modifications in line 459: "heave ( $A_H = 0.03D$ ;  $St_H = 0.09$ ), surge ( $A_S = 0.06D$ ;  $St_S = 0.11$ ) and pitch ( $A_{P0.14} = 4^\circ$ ;  $St_{P0.14} = 0.14$  and  $A_{P0.28} = 4^\circ$ ;  $St_{P0.28} = 0.28$ ) motions".

 Is it correct that the results in this paper find an amplification of wake displacement at lower Strouhal numbers (heave St=0.09 and surge St=0.11) than those for which Messmer et al. (2024) found optimal wake recovery (St=0.3-0.6)? A small note or discussion about that would be interesting.

Indeed, in this paper the authors wanted to focus the study on realistic motions (secondorder motions due to the response of the floater linked to mooring lines and anchoring characteristics), and not on optimal impact ones. Thus, the latter would provide higher wake parameter amplifications than those find here.

Added to the article line 492: "The Strouhal numbers of the investigated motions are lower than those of maximal impact found in previous studies (Li et al., 2022; Messmer et al., 2024), one expects that applying such similar Strouhal numbers to our configuration would provide higher wake parameter modifications".

line 511: 'wake dynamics are' instead of 'is'

Modifications in line 496: "are".

• line 517 to line 522: This is a very interesting statement, and some thoughts come to the mind: The work of Bossuyt et al. (Floating wind farm experiments through scaling for wake characterization, power extraction, and turbine dynamics, 2023) studied floating turbine models that can move freely in a water-wind tunnel experiment. Thus, the turbines moved with a spectrum of frequencies and with all degrees of freedom simultaneously. Upon checking, the strouhal numbers seem similar to results here, but limited to 0.13, while the amplitudes are likely much larger. Severe wake displacements were registered in that work, though no comparison with static misalignment models was made. Checking if the results of Bossuyt et al. match the conclusions in this paper and confirm an amplification would give very interesting and helpful context about the sensitivity of amplification to single frequency motion. It is an important note to make in the context of this paper.

Indeed, Bossuyt et al. (2023) found that wave conditions triggering pitch motions induce vertical motions of the wake centre, and that associated to yaw motions induce horizontal ones, which are consistent with the results found in the present paper. However, they experimented free-motion turbine models subjected to harmonic wave conditions, which results to approximately harmonic turbine motions (Fig. 19), closer to that is performed in

the present paper than to the actual range of amplitudes and frequencies of full-scale motions. Indeed, regarding Schliffke et al. (2024) (Fig. 2), the motion spectra of the full-scale wind turbine do not present any specific frequency peak.